# LayoutDETR: Detection Transformer Is a Good Multimodal Layout Designer

## Abstract

Graphic layout designs play an essential role in visual communication. Yet hand-crafting layout designs is skill-demanding, time-consuming, and non-scalable to batch production. Generative models emerge to make design automation scalable but it remains non-trivial to produce designs that comply with designers' multimodal desires, i.e., constrained by background images and driven by foreground content. We propose *LayoutDETR* that inherits the high quality and realism from generative modeling, while reformulating content-aware requirements as a detection problem: we learn to detect in a background image the reasonable locations, scales, and spatial relations for multimodal foreground elements in a layout. Our solution sets a new state-of-the-art performance for layout generation on public benchmarks and on our newly-curated ad banner dataset. We integrate our solution into a graphical system that facilitates user studies, and show that users prefer our designs over baselines by significant margins.

## 1 Introduction

Graphic layout designs are at the foundation of communication between media designers and their target audience (Landa, 2010; Lok & Feiner, 2001; Stribley, 2016). Multimodal elements, i.e., foreground images/texts, are framed by layout bounding boxes and reasonably arranged on a background image. This relies on a thoughtful understanding of the semantics of each element and their harmony as a whole. Therefore, handcrafting such layout designs is skill-demanding, time-consuming, and requires experienced professionals. In practice, it has been impossible to batch-produce them in massive quantities (Chen et al., 2019).

Growing demands of automating graphic layout designs have motivated researchers to adapt deep generative models (Goodfellow et al., 2014; Kingma & Welling, 2013; Larsen et al., 2016; Chen et al., 2018; Rombach et al., 2022) to this task (Kikuchi et al., 2021; Zheng et al., 2019; Patil et al., 2020; Guo et al., 2021; Gupta et al., 2021; Zhou et al., 2022; Cao et al., 2022; Cheng et al., 2023; Hsu et al., 2023), but few of them investigate generation conditioned on multimodal inputs and their fusion for layout designs.

Conditioning on multimodal inputs is critical to enrich designers' control and to command the aesthetics of layout designs. In this paper, we focus on them. We propose to equip designers with an AI-empowered system that allows multimodal input: arbitrary background images, foreground images, and foreground copywriting texts from varying categories. Fig. 1 depicts the functionality and resulting design samples of our solution. We learn a generative model of conditional layout distribution that is constrained by background images and driven by foreground elements. This requires our model to (1) learn the prior distribution from large-scale realistic layout samples, (2) understand the appearance and semantics of background images, (3) understand the appearance and semantics of foreground elements, and (4) fuse background and foreground information to generate the layout bounding box parameters of each foreground element.

To tackle (1), we inherit and explore the high realism from three types of generative learning paradigms: generative adversarial networks (GANs) (Goodfellow et al., 2014; Karras et al., 2020), variational autoencoders (VAEs) (Kingma & Welling, 2013), and VAE-GAN (Larsen et al., 2016). To handle (2), we reformulate the background conditioned layout generation as a detection problem, considering both problems require visual understanding and optimize for bounding box parameters. We integrate these two seemingly irrelevant techniques and train a DETR-flavored detector (Carion et al., 2020) as a layout generator. Specifically, we employ the visual transformer encoder and bounding box transformer decoder architectures of DETR and jointly optimize its supervision loss

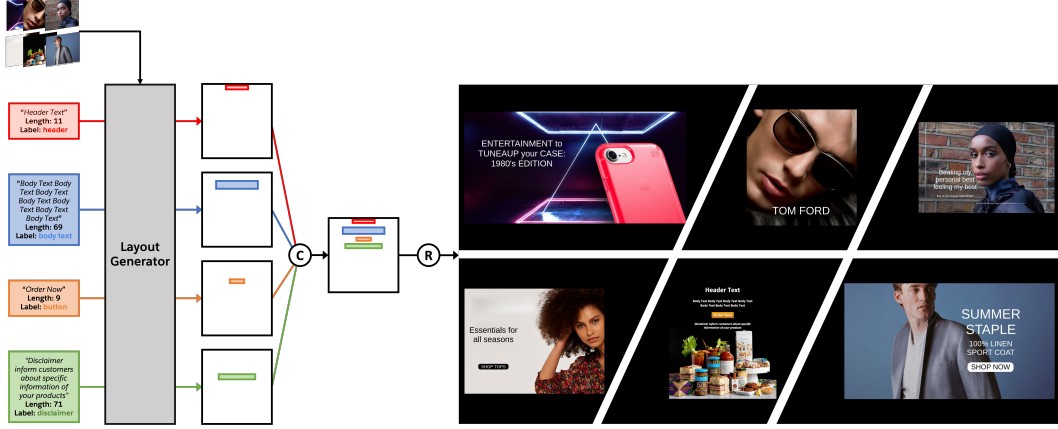

Figure 1: **Left**: LayoutDETR takes a background image and a set of multimodal foreground elements (images/texts) as input, and outputs an aesthetically appealing layout. **Right**: we show a few banner samples with rendered texts using our auto-designed layouts. "C" is the composition and "R" the rendering process.

with generative adversarial loss. We hence name our solution *LayoutDETR*. To handle (3), we incorporate a CNN-based image encoder (Caron et al., 2020) and a BERT-based text encoder (Devlin et al., 2019) for the multimodal foreground inputs, and feed the features as input tokens to DETR transformer decoder. (4) is naturally handled by DETR thanks to its transformer nature: foreground elements interact with each other through the self-attention in the decoder, while background features interact through the cross-attention between the image encoder and layout decoder.

We summarize our **contributions** as: **(1) Method**. We bridge two seemingly irrelevant research domains, layout generation and visual detection, into one framework that solves multimodal graphic layout design constrained by background images and driven by foreground image/text elements. No existing methods can handle all these modalities at once. **(2) Dataset**. We establish a large-scale ad banner dataset with rich semantic annotations including text bounding boxes, text categories, and text content. We benchmark this dataset for graphic layout generation over six recent methods, three of our solution variants, and conduct ablation study. We will release the dataset. **(3) State-of-the-art performance**. Our solution reaches a new state-of-the-art performance for graphic layout generation in a comprehensive set of six evaluation metrics, which measure the realism, accuracy, and regularity of generated layouts. **(4) Graphical system and user study**. We integrate our solution into a graphical system that scales up layout generation and facilitates user studies. Users prefer our designs over all baselines by significant margins.

## 2 RELATED WORK

**Deep generative models for layout design.** Automating the layout design with high quality and realism gains increasing attention and achieves substantial progress, thanks to the revolutionary advancements of deep generative models, especially generative adversarial networks (GANs) (Goodfellow et al., 2014; Brock et al., 2019; Karras et al., 2020), variational autoencoders (VAEs) (Kingma & Welling, 2013; Larsen et al., 2016), autoregressive models (ARMs) (Van den Oord et al., 2016; Salimans et al., 2017; Chen et al., 2018), and diffusion models (Ho et al., 2020; Song et al., 2021; Rombach et al., 2022). Researchers adopt generative models to learn to generate bounding box parameters, in the form of center location, height, width, and optionally depth, for each foreground element in a layout in either graphics or natural scene domains. Orthogonal to learning paradigms, existing methods also investigate a variety of generator architectures including multi-layer perceptron (MLP) (Krizhevsky et al., 2012), convolutional neural networks (CNN) (Simonyan & Zisserman, 2015), recursive neural networks (RvNN) (Socher et al., 2013), long short-term memory (LSTM) (Haykin & Network, 2004), graph convolutional networks (GCN) (Kipf & Welling, 2017), transformer (Vaswani et al., 2017), etc. Some layout design methods (Kikuchi et al., 2021; Zheng et al., 2019; Patil et al., 2020; Guo et al., 2021; Gupta et al., 2021; Zhou et al., 2022; Cao et al., 2022; Hsu et al., 2023; Cheng et al., 2023; Levi et al., 2023; Zhang et al., 2023; Lin et al., 2023) focus on graphics domain only, while others (Jyothi et al., 2019; Gupta et al., 2021; Tan et al., 2018; Zhao et al., 2018; Lee et al., 2018) generalize to natural scenes or even 3D data. Some methods (Li et al., 2019; Kikuchi et al., 2021; Patil et al., 2020; Gupta et al., 2021; Kong et al., 2022; Levi et al., 2023; Zhang et al., 2023) allow only bounding box category condition, while others (Li et al., 2020; Lee et al., 2020; Zheng et al., 2019; Guo et al., 2021; Zhou et al., 2022; Cao et al., 2022; Hsu et al., 2023; Cheng et al., 2023) allow richer conditions in multi-modalities in the form of background

Table 1: A taxonomy of existing and our layout generation methods. We tag for each method its working data domain(s), data modality(ies) of input condition(s), backend generative model type, as well as implementation architecture. Our method and its variants are the only ones that enable full control of varying multimodal conditions and integrate object detection techniques into layout generation.

| Method | Domain(s) | Conditioning modality | Generative model | Architecture |
|---|---|---|---|---|
| LayoutGAN (Li et al., 2019) | Graphics | Bbox category | GAN | MLP, CNN |
| LayoutGAN+ (Li et al., 2020) | Graphics | Bbox category, attribute vector | GAN | MLP, CNN |
| LayoutGAN++ (Kikuchi et al., 2021) | Graphics | Bbox category | GAN | Transformer |
| DS-GAN (Hsu et al., 2023) | Graphics | Bbox category, bg images | GAN | CNN, LSTM |
| House-GAN (Nauata et al., 2020) | Floor plans | Bubble diagram | GAN | CNN, MPN |
| House-GAN++ (Nauata et al., 2021) | Floor plans | Bubble diagram | GAN | CNN, MPN |
| LayoutVAE (Jyothi et al., 2019) | Natural scenes | Bbox category | VAE | MLP, LSTM |
| READ (Patil et al., 2020) | Graphics | Bbox category | VAE | RvNN |
| Vinci (Guo et al., 2021) | Graphics | Bbox category, bg/fg images, text | VAE | LSTM |
| NDN (Lee et al., 2020) | Graphics | Bbox category, spatial relation | VAE | GCN |
| VTN (Arroyo et al., 2021) | Graphics, natural scenes | Bbox category | VAE | Transformer |
| CanvasVAE (Yamaguchi, 2021) | Graphics | Bbox category, attribute vector | VAE | Transformer |
| C2F-VAE (Jiang et al., 2022b) | Graphics | Bbox category | VAE | Transformer |
| DeepSVG (Carlier et al., 2020) | Vector graphics | Vector paths, attribute vector, command | VAE | Transformer |
| ContentGAN (Zheng et al., 2019) | Graphics | Bbox category, fg image, text | VAE + GAN | Transformer |
| TextLogoLayout (Wang et al., 2022) | Graphics | Glyph | VAE + GAN | RNN, CNN |
| LayoutMCL (Nguyen et al., 2021) | Graphics | Bbox category | ARM | RNN, CNN |
| CanvasEmb (Xie et al., 2021) | Graphics | Bbox category, attribute vector | ARM | Transformer |
| LayoutTransformer (Gupta et al., 2021) | Graphics, natural scenes, 3D | Bbox category | ARM | Transformer |
| BLT (Kong et al., 2022) | Graphics, natural scenes, 3D | Bbox category | ARM | Transformer |
| UniLayout (Jiang et al., 2022a) | Graphics | Bbox category | ARM | Transformer |
| Parse-Then-Place (Lin et al., 2023) | Graphics | Text | ARM | LLM, Transformer |
| LayoutFormer++ (Jiang et al., 2023) | Graphics | Bbox category, spatial relation | ARM | Transformer |
| PLay (Cheng et al., 2023) | Graphics | Bbox category, guideline | Diffusion | Transformer |
| DLT (Levi et al., 2023) | Graphics | Bbox category | Diffusion | Transformer |
| LayoutDiffusion (Zhang et al., 2023) | Graphics | Bbox category | Diffusion | Transformer |
| CGL-GAN (Zhou et al., 2022) | Graphics | Bg image | GAN + DETR | Transformer |
| ICVT (Cao et al., 2022) | Graphics | Bg image | VAE + DETR | Transformer |
| LayoutDETR-GAN (ours) | Graphics | Bbox category, bg/fg images, text | GAN + DETR | Transformer |
| LayoutDETR-VAE (ours) | Graphics | Bbox category, bg/fg images, text | VAE + DETR | Transformer |
| LayoutDETR-VAE-GAN (ours) | Graphics | Bbox category, bg/fg images, text | VAE + GAN + DETR | Transformer |

images, foreground images, or foreground texts, but not all. A comprehensive taxonomy of layout generation methods is in Table 1.

None of the existing multimodal layout generation methods is designed to handle all the background and foreground modalities at once: LayoutGAN+ (Li et al., 2020) and NDN (Lee et al., 2020) are unaware of background and foreground image elements in their formulation. CGL-GAN (Zhou et al., 2022), ICVT (Cao et al., 2022), and DS-GAN (Hsu et al., 2023) are unaware of foreground text and image elements in their formulation. ContentGAN (Zheng et al., 2019) does not take layout spatial information for training and does not even use complete background images as input during training. This does not benefit the layout representation, layout regularity, or final quality (e.g., text readability) after rendering foreground elements onto the background. Vinci (Guo et al., 2021) relies on a finite set of predefined layout candidates to choose background images from a pool of food and beverage domains. Their method is unable to design layouts conditioned on arbitrary backgrounds in open domains, natural or handcrafted, plain or cluttered, like ours. Considering multimodal layout design persists as a challenging problem, we focus our scope on graphic layouts: mobile application UIs and our newly-organized large-scale ad banners.

**Object detection.** The goal of object detection is to predict a set of bounding boxes and their categories for each object of interest in a query image. Modern detectors address this in one of three ways: Two-stage detectors (Girshick, 2015; He et al., 2017) predict boxes based on proposals; single-stage detectors predict boxes based on anchors (Lin et al., 2017) or a grid of possible object centers (Redmon et al., 2016); direct detectors (Carion et al., 2020; Dai et al., 2021) avoid those hand-crafted initial guesses and directly predict absolute parameters of boxes in the query images. Direct detectors (Carion et al., 2020; Dai et al., 2021) inspire us to leverage their powerful image understanding capability, and combine it with modern layout generators, in order to address the problem of multimodal layout generation. Similar in spirit to CGL-GAN (Zhou et al., 2022) or ICVT (Cao et al., 2022), we take advantage of the transformer encoder-decoder architecture and generalized intersection over union (gIoU) loss (Rezatofighi et al., 2019) in DETR to learn layout distributions from background image contexts. Unlike prior art, our layout outputs are additionally conditioned on foreground image/text elements.

## 3 LAYOUTDETR

**Problem statement.** A graphic layout sample $\mathcal{L}$ is represented by a set of $N$ 2D bounding boxes $\{\mathbf{b}^i\}_{i=1}^N$. Each $\mathbf{b}^i$ is parameterized by a vector with four elements: its center location in a background image $(y^i, x^i)$, height $h^i$, and width $w^i$. In order to handle background images $\mathbf{B}$ with arbitrary sizes $(H, W)$, we normalize box parameters by their image size correspondingly, i.e., $\mathcal{L} = \{(y^i/H, x^i/W, h^i/H, w^i/W)\}_{i=1}^N \doteq \{\hat{\mathbf{b}}^i\}_{i=1}^N$. The multimodal inputs are the

background image $\mathbf{B}$ and a set of $N$ foreground elements, in the form of either text elements $\mathcal{T} = \{\mathbf{t}^i\}_{i=1}^M = \{(\mathbf{s}^i, c^i, l^i)\}_{i=1}^M$ or image patches $\mathcal{P} = \{\mathbf{p}^i\}_{i=1}^K$, where $M \geq 0$, $K \geq 0$, $M + K = N$. $\mathbf{s}^i$ is a text string, $c^i$ is the text class from the set {*header text*, *body text*, *disclaimer text*, *button text*}, and $l^i$ is the length of the text string. Each foreground element corresponds to a bounding box in the layout, indicating its location and size. In case there are foreground elements that we are not interested in, e.g., button underlays or embellishments, we leave them as part of the background. Our goal is to learn a layout generator $G$ that takes a latent noise vector $\mathbf{z}$ and the multimodal conditions as input, and outputs a realistic and reasonable layout complying with the multimodal control: $G(\mathbf{z}, \mathbf{B}, \mathcal{T} \cup \mathcal{P}) \mapsto \mathcal{L}_{\text{fake}}$.

## 3.1 GENERATIVE LEARNING FRAMEWORKS

**GAN variant.** Following the GAN paradigm (Goodfellow et al., 2014; Brock et al., 2019; Karras et al., 2018; 2019; 2020; Yu et al., 2020; 2021; Sauer et al., 2022; Lee et al., 2022; Yu et al., 2022), the generator $G$ is simultaneously and adversarially trained against discriminator $D$ training. We formulate a multimodal-conditional discriminator $D^c(\mathcal{L}, \mathbf{B}, \mathcal{T} \cup \mathcal{P}) \mapsto \{0, 1\}$, as well as an unconditional discriminator $D^u(\mathcal{L}) \mapsto \{0, 1\}$.

It has been observed that the discriminators are insensitive to irregular bounding box positions (Kikuchi et al., 2021). For example, the discriminators tend to overlook the unusual layout where a *header texts* is placed at the bottom. As a result, we follow the self-learning technique in (Liu et al., 2020) and add position-aware regularization to our discriminators. Similar to (Kikuchi et al., 2021), we add an auxiliary decoder to each discriminator to reconstruct its input. The decoders $F^c/F^u$ take the output features $\mathbf{f}^c/\mathbf{f}^u$ of their discriminators $D^c/D^u$, add them with learnable positional embeddings $\mathcal{E}^c = \{\mathbf{e}_i^c\}_{i=1}^N$ / $\mathcal{E}^u = \{\mathbf{e}^u\}_{i=1}^N$, and reconstruct the bounding box parameters and multimodal conditions that are the input of the discriminators: $F^c(\mathbf{f}^c, \mathcal{E}^c) \mapsto \mathcal{L}_{\text{dec}}^c, \mathbf{B}_{\text{dec}}, \mathcal{T}_{\text{dec}} \cup \mathcal{P}_{\text{dec}}$ / $F^u(\mathbf{f}^u, \mathcal{E}^u) \mapsto \mathcal{L}_{\text{dec}}^u$. Using $\mathcal{E}^c/\mathcal{E}^u$ is necessary, without which the reconstructed bounding boxes in a layout would have no variance as they would be conditioned on identical features. The decoders are jointly trained with the discriminators to minimize the reconstruction loss. It enforces the discriminators to fully condition on all their inputs for the binary classification. See Fig. 2 Yellow for the diagram. Thus, our adversarial learning objective is:

$$\min_G \max_{D^c, F^c, \mathcal{E}^c, D^u, F^u, \mathcal{E}^u} L_{\text{GAN}} = L_{\text{GAN\_fake}} + L_{\text{GAN\_real}} \tag{1}$$

$$L_{\text{GAN\_fake}} \doteq \mathbb{E}_{\mathbf{z} \sim \mathcal{N}(\mathbf{0}, \mathbf{I}), \{\mathbf{B}, \mathcal{T} \cup \mathcal{P}\} \sim P_{\text{data}}} - \log D^c(\mathcal{L}_{\text{fake}}, \mathbf{B}, \mathcal{T} \cup \mathcal{P}) - \log D^u(\mathcal{L}_{\text{fake}}) \tag{2}$$

$$L_{\text{GAN\_real}} \doteq \mathbb{E}_{\{\mathcal{L}_{\text{real}}, \mathbf{B}, \mathcal{T} \cup \mathcal{P}\} \sim P_{\text{data}}} \log D^c(\mathcal{L}_{\text{real}}, \mathbf{B}, \mathcal{T} \cup \mathcal{P}) + \log D^u(\mathcal{L}_{\text{real}}) - L_{\text{dec}} \tag{3}$$

$$L_{\text{dec}} = \lambda_{\text{layout}} \big( L_{\text{layout}}(\mathcal{L}_{\text{dec}}^c, \mathcal{L}_{\text{real}}) + L_{\text{layout}}(\mathcal{L}_{\text{dec}}^u, \mathcal{L}_{\text{real}}) \big) + \lambda_{\text{im}} \big( ||\mathbf{B}_{\text{dec}} - \mathbf{B}||_2 + L_{\text{im}}(\mathcal{P}_{\text{dec}}, \mathcal{P}) \big) + L_{\text{text}}(\mathcal{T}_{\text{dec}}, \mathcal{T}) \tag{4}$$

$$L_{\text{layout}}(\mathcal{L}_1, \mathcal{L}_2) = \frac{1}{N} \sum_{i=1}^N ||\hat{\mathbf{b}}_1^i - \hat{\mathbf{b}}_2^i||_2 \tag{5}$$

$$L_{\text{im}}(\mathcal{P}_1, \mathcal{P}_2) = \frac{1}{N} \sum_{i=1}^N ||\mathbf{p}_i^1 - \mathbf{p}_2^i||_2 \tag{6}$$

$$L_{\text{text}}(\mathcal{T}_1, \mathcal{T}_2) = \frac{1}{N} \sum_{i=1}^N \big( \lambda_{\text{str}} L_{\text{str}}(\mathbf{s}_1, \mathbf{s}_2) + \lambda_{\text{cls}} L_{\text{cls}}(c_1, c_2) + \lambda_{\text{len}} L_{\text{len}}(l_1, l_2) \big) \tag{7}$$

where $L_{\text{str}}$, $L_{\text{cls}}$, $L_{\text{len}}$ are the reconstruction losses for foreground text strings, text classes, and text lengths respectively. $L_{\text{str}}$ is the auto-regressive loss according to BERT language modeling (Devlin et al., 2019; Li et al., 2022b). $L_{\text{cls}}$ is the standard classification cross-entropy loss. So is $L_{\text{len}}$, considering we quantize and classify a string length integer into one of 256 levels $[0, 255]$. $\lambda_{\text{layout}} = 500.0$, $\lambda_{\text{im}} = 0.5$, $\lambda_{\text{str}} = 0.1$, $\lambda_{\text{cls}} = 50.0$, and $\lambda_{\text{len}} = 2.0$.

**VAE variant.** VAEs are an alternative paradigm to GANs for generative models. Following the VAE paradigm (Kingma & Welling, 2013; Rezende et al., 2014), the generator $G$ is jointly trained with an encoder $E$ that maps from the layout space to the latent noise distribution space. The output of $E$ are the mean $\mu$ and covariance matrix $\Sigma$ of a multivariate Gaussian distribution, $E(\mathcal{L}) \mapsto \mu, \Sigma$, the samples of which are input to $G$: $\text{sample}(\mu, \Sigma) = \mathbf{z}_0^T \Sigma^{\frac{1}{2}} \mathbf{z}_0 + \mu$, $\mathbf{z}_0 \sim \mathcal{N}(\mathbf{0}, \mathbf{I})$. It represents the differentiable reparameterization trick in the standard VAE pipeline (Kingma & Welling, 2013; Rezende et al., 2014). See Fig. 2 Blue.

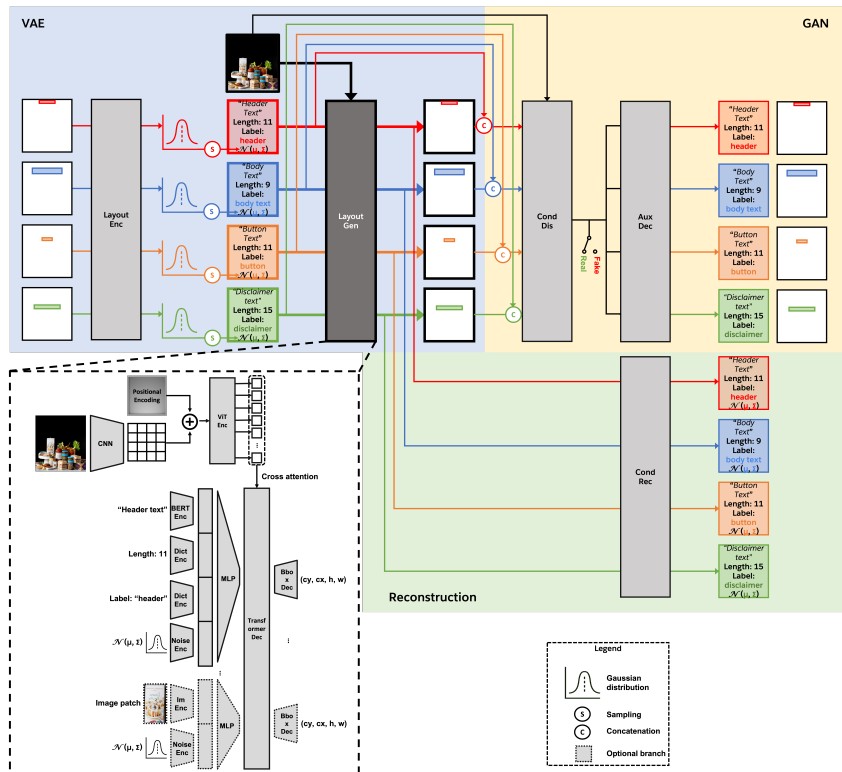

Figure 2: Our unified training framework covers three generator variants: GAN-, VAE-, and VAE-GAN-based. The layout generator network (darker color and bold) appears in all variants. Its DETR-based multimodal architecture is at the bottom left. During inference, only the generator is needed.

The conditional VAE minimizes the reconstruction loss:

$$\min_{E,G} L_{\text{VAE}} \doteq \mathbb{E}_{\{\mathcal{L}_{\text{real}}, \mathbf{B}, \mathcal{T} \cup \mathcal{P}\} \sim P_{\text{data}}} \lambda_{\text{layout}} L_{\text{layout}}(\mathcal{L}_{\text{fake}}, \mathcal{L}_{\text{real}}) + \lambda_{\text{KL}} \text{KL}\big(E(\mathcal{L}_{\text{real}})||\mathcal{N}(\mathbf{0}, \mathbf{I})\big) \tag{8}$$

$\text{KL}(\cdot||\cdot)$ is the Kullback-Leibler (KL) Divergence used in standard VAE (Kingma & Welling, 2013; Rezende et al., 2014) to regularize the encoded latent noise distribution. The hyper-parameters $\lambda_{\text{layout}} = 500.0$ and $\lambda_{\text{KL}} = 1.0$.

**VAE-GAN variant.** VAEs and GANs are also compatible with each other. See Fig. 2 Blue plus Yellow. Following (Larsen et al., 2016; Zheng et al., 2019), we jointly optimize Eq. 1 and 8:

$$\min_{E,G} \max_{D^c, F^c, \mathcal{E}^c, D^u, F^u, \mathcal{E}^u} L_{\text{GAN}} + L_{\text{VAE}} \tag{9}$$

### 3.2 ADDITIONAL OBJECTIVES

Other losses and regularization terms also play important roles in the generated layout quality. First, we add bounding box supervision as in DETR (Carion et al., 2020). We use the generalized intersection over union loss $\text{gIoU}(\cdot, \cdot)$ (Rezatofighi et al., 2019) between generated layout and its ground truth $L_{\text{gIoU}}(\mathcal{L}_{\text{fake}}, \mathcal{L}_{\text{real}})$, where:

$$L_{\text{gIoU}}(\mathcal{L}_1, \mathcal{L}_2) \doteq \lambda_{\text{gIoU}} \frac{1}{N} \sum_{i=1}^{N} \text{gIoU}(\hat{\mathbf{b}}_1^i, \hat{\mathbf{b}}_2^i) \tag{10}$$

where the hyper-parameter $\lambda_{\text{gIoU}} = 4.0$.

Second, we introduce an auxiliary reconstructor $R$ for the generator to enhance the controllability of input conditions. $R$ takes the last features of $G$, $\mathcal{F} = \{\mathbf{f}_i^g\}_{i=1}^N$, as input tokens before outputting box parameters, and learns to reconstruct $\mathcal{P}$ and $\mathcal{T}$: $R(\mathcal{F}) \mapsto \mathcal{P}_{\text{rec}}, \mathcal{T}_{\text{rec}}$.

Jointly with $G$ training, we learn to minimize:

$$L_{\text{rec}} \doteq \lambda_{\text{im}} L_{\text{im}}(\mathcal{P}_{\text{rec}}, \mathcal{P}) + L_{\text{text}}(\mathcal{T}_{\text{rec}}, \mathcal{T}) \tag{11}$$

with $L_{\text{im}}$ from Eq. 6 and $L_{\text{text}}$ from 7. See Fig. 2 Green.

Third, reasonable layout designs typically avoid overlapping between foreground elements. We leverage the overlap loss $L_{\text{overlap}} \doteq \lambda_{\text{overlap}} L_{\text{overlap}}(\mathcal{L}_{\text{fake}})$ from (Li et al., 2020) that discourages overlapping between any pair of bounding boxes in a generated layout. We set $\lambda_{\text{overlap}} = 7.0$.

Fourth, aesthetically appealing layouts usually maintain one of the six alignments between a pair of adjacent bounding boxes: left, horizontal-center, right, top, vertical-center, and bottom aligned. We leverage the misalignment loss $L_{\text{misalign}} \doteq \lambda_{\text{misalign}} L_{\text{misalign}}(\mathcal{L}_{\text{fake}})$ follow (Li et al., 2020) that discourages misalignment. We set $\lambda_{\text{misalign}} = 17.0$.

Finally, our training objective is formulated as follows.

$$\min_{E,G,R} \quad \max_{D^c,F^c,\mathcal{E}^c,D^u,F^u,\mathcal{E}^u} L_{\text{GAN}} + L_{\text{VAE}} + L_{\text{gIoU}} + L_{\text{rec}} + L_{\text{overlap}} + L_{\text{misalign}} \tag{12}$$

All the $\lambda$s are trivially set to align the order of magnitude of each loss term. We use an identical set of $\lambda$s for all the datasets to validate our performance is insensitive to $\lambda$s.

### 3.3 DETR-based Multimodal Architectures

Architecture design is where we integrate object detection with layout generation. Detection transformer (DETR) (Carion et al., 2020) is employed and modified for LayoutDETR generator $G$ and conditional discriminator $D^c$.

As depicted in Fig. 2 bottom left, $G$ and $D^c$ contain a background encoder and a layout transformer decoder. The encoder is the same as in DETR (Carion et al., 2020). The decoder is inherited from the DETR decoder with self-attention and encoder-decoder-cross-attention mechanisms (Vaswani et al., 2017). Different from DETR, we replace their freely-learnable object queries with our foreground embeddings as the input tokens to the decoder. The decoder then transforms the embeddings to layout bounding box parameters. Foreground embedding is a concatenation of noise embedding and text/image embedding. Text embedding is a concatenation of text string embedding (BERT-pretrained and fixed), text class embedding (via learning dictionary), and text length embedding (via learning dictionary). See more clarifications on our design choices in Section A in Appendix. See the implementations of the other networks in Section B in Appendix. In particular, other networks are transformer-based as well. Only the input and output formats differ depending on the network functionality. We do not choose a wire-frame discriminator since its empirical performance is worse, as also observed in (Kikuchi et al., 2021).

## 4 New Ad Banner Dataset

Not all existing datasets are suitable for multimodal layout design because they do not always provide multi-modality information, e.g. natural scene datasets or Crello graphic document dataset (Yamaguchi, 2021). Other datasets, such as PubLayNet document dataset (Zhong et al., 2019) and PartNet 3D shape dataset (Yu et al., 2019) render layouts only on plain background. Magazine dataset (Zheng et al., 2019) does not provide complete background images since they have masked out foreground layouts from the background. On the other hand, ad banner datasets are composed of multimodal elements and lead to several previous layout design techniques (Guo et al., 2021; Zhou et al., 2022; Cao et al., 2022; Hsu et al., 2023). Unfortunately, none of their datasets is publicly available, except for CGL dataset (Zhou et al., 2022), PKU PosterLayout (Hsu et al., 2023), Text-LogoLayout dataset (Wang et al., 2022) which, however, do not contain the widely used English modality. We therefore collect a new large-scale ad banner dataset of 7,196 samples containing English characters, which are ready to release.

Each of our samples consists of a well-designed banner image, its layout ground truth, foreground text strings, text classes, and background image. The banner images are filtered from Pitt Image Ads Dataset (Hussain et al., 2017) and crawled from Google Image Search Engine. Their layouts and text classes are manually annotated by Amazon Mechanical Turk (AMT). The text classes are {*header text*, *body text*, *disclaimer text*, *button text*}, with logos categorized as *header text*. The text strings are extracted by OCR (pad) and removed by image inpainting (Suvorov et al., 2022) to obtain the text-free background image. Examples and more details about data collection are in Section C and Fig. 4-5 in Appendix.

## 5 Experiments

**Datasets.** Ablation study and user study are conducted on our ad banner dataset with 7,196 samples. Comparisons to baselines are additionally performed on the CGL dataset with 59,978 valid Chinese ad banner samples (Zhou et al., 2022), and on the CLAY dataset with 32,063 valid mobile application UI samples (Li et al., 2022a). We apply the same OCR and image inpainting processes to CGL and CLAY datasets, in order to extract texts as part of input conditions, and separate apart foreground from background. 90% of the samples are used for training and 10% for testing.

Table 2: Ablation study w.r.t. loss config (top) or conditional embedding config (bottom) on **our ad banner dataset**. Each row is progressively ablated from its row above. Each cell contains mean±std. For FIDs and KIDs, they are the statistics over 10 runs. ⇑/⇓ indicates a higher/lower value is better. **Bold**/underline font indicates the top/second best value in the column.

| Method | Realism | | | | Accuracy | | Regularity | |
|---|---|---|---|---|---|---|---|---|
| | Layout FID ⇓ | Layout KID $(\times 10^{-3})$ ⇓ | Image FID ⇓ | Image KID $(\times 10^{-5})$ ⇓ | IoU ⇑ | DocSim ⇑ | Overlap ⇓ | Misalign $(\times 10^{-2})$ ⇓ |
| Random layout | $58.21_{\pm 4.04}$ | $525.93_{\pm 45.08}$ | $51.01_{\pm 0.41}$ | $582.47_{\pm 7.53}$ | – | – | – | – |
| Conditional LayoutGAN++ | $11.33_{\pm 0.10}$ | $44.77_{\pm 0.36}$ | $36.06_{\pm 0.02}$ | $115.16_{\pm 3.37}$ | $0.111_{\pm 0.001}$ | $0.121_{\pm 0.001}$ | $0.374_{\pm 0.006}$ | $2.194_{\pm 0.058}$ |
| + Aux. Dec. (Eq. 4-7) | $4.25_{\pm 0.01}$ | $16.62_{\pm 0.05}$ | $28.40_{\pm 0.06}$ | $58.5_{\pm 1.45}$ | $0.163_{\pm 0.002}$ | $0.130_{\pm 0.001}$ | $0.104_{\pm 0.003}$ | $0.759_{\pm 0.021}$ |
| + Gen. Rec. (Eq. 11) | $3.27_{\pm 0.01}$ | $11.80_{\pm 0.04}$ | $29.56_{\pm 0.06}$ | $11.29_{\pm 0.20}$ | $0.186_{\pm 0.002}$ | $0.148_{\pm 0.001}$ | $0.125_{\pm 0.003}$ | $0.853_{\pm 0.016}$ |
| + Uncond. Dis. $D^{\mathrm{u}}$ | $3.70_{\pm 0.05}$ | $16.23_{\pm 0.08}$ | $29.21_{\pm 0.08}$ | $25.09_{\pm 0.02}$ | $0.177_{\pm 0.002}$ | $0.140_{\pm 0.001}$ | $0.103_{\pm 0.003}$ | $0.681_{\pm 0.011}$ |
| + gIoU loss (Eq. 10) | $3.23_{\pm 0.01}$ | $11.60_{\pm 0.02}$ | $28.20_{\pm 0.04}$ | $10.51_{\pm 0.09}$ | $0.182_{\pm 0.002}$ | $0.138_{\pm 0.001}$ | $0.106_{\pm 0.003}$ | $0.721_{\pm 0.011}$ |
| + Overlap & Misalign loss $\doteq$ LayoutDETR-GAN (ours) | **$3.19_{\pm 0.01}$** | **$5.62_{\pm 0.01}$** | **$27.35_{\pm 0.04}$** | **$8.31_{\pm 0.80}$** | **$0.208_{\pm 0.002}$** | **$0.151_{\pm 0.000}$** | **$0.101_{\pm 0.003}$** | **$0.646_{\pm 0.011}$** |
| - Text length embeddings | $3.24_{\pm 0.01}$ | $9.25_{\pm 0.05}$ | $28.65_{\pm 0.03}$ | $11.42_{\pm 0.35}$ | $0.191_{\pm 0.002}$ | $0.144_{\pm 0.001}$ | $0.117_{\pm 0.003}$ | $0.807_{\pm 0.012}$ |
| - Text class embeddings | $25.17_{\pm 0.54}$ | $171.88_{\pm 5.17}$ | $29.25_{\pm 0.25}$ | $139.16_{\pm 4.44}$ | $0.166_{\pm 0.002}$ | $0.132_{\pm 0.001}$ | $0.110_{\pm 0.001}$ | $0.000_{\pm 0.000}$ |

**Baselines.** We select six recent methods covering a variety of generative paradigms and architectures listed in Table 1: LayoutGAN++ (Kikuchi et al., 2021), READ (Patil et al., 2020), Vinci (Guo et al., 2021), LayoutTransformer (Gupta et al., 2021), CGL-GAN (Zhou et al., 2022), and ICVT (Cao et al., 2022). We noted that LayoutTransformer (Gupta et al., 2021) is empirically superior to the more recent work BLT (Kong et al., 2022), the same observation as Row 1 v.s. Row 8 in Table 2 of (Jiang et al., 2022a). We therefore did not experiment with BLT. Several baselines do not allow background conditions and/or foreground text/image conditions. We integrate our encoders to their models for fair comparisons.

**Evaluation metrics.** (1) For the **realism** of generated layouts, we calculate the Fréchet distances (Heusel et al., 2017) and kernel distances (Bińkowski et al., 2018) between fake and real feature distributions. All the real testing samples are used, and the same number of generated samples are used. We consider two feature spaces: the layout features pretrained by (Kikuchi et al., 2021), and VGG image features pretrained on ImageNet (Heusel et al., 2017). We obtain the output banner images by overlaying foreground image patches and rendering foreground texts on top of background images according to the generated layout. The rendering process and examples are in Fig. 3. (2) For the sample-wise **accuracy** of generated layouts w.r.t. their ground truth, we calculate the layout-to-layout IoU (Kikuchi et al., 2021) and DocSim (Patil et al., 2020). Box-level matching is trivial as the correspondences between generated and ground truth boxes are given by the conditioning input. (3) For the **regularity** of generated layouts, our metrics are the overlap loss (Li et al., 2020) and misalignment loss (Li et al., 2020) in Section 3.2.

### 5.1 ABLATION STUDY

**Loss configurations.** We start from the LayoutGAN++ (Kikuchi et al., 2021) baseline implementation and additionally enable it to take multimodal foreground and background as input conditions. We then progressively add on extra loss terms and report the quantitative measurements in Table 2 top section. We observe: (1) Row 1 contains the far worse results of randomly generated layouts on real backgrounds, indicating that the quality of layouts itself matters. (2) Comparing Row 2 and 3, all the metrics are improved, as the auxiliary decoder enhances the discriminator's conditioning and representation. (3) Comparing Row 3 and 4, all the metrics are improved, thanks to the enhanced controllability through the reconstruction of conditional inputs. (4) Comparing Row 4 and 5, unconditional discriminator benefits the layout regularity due to its approximation power between generated layout parameters and real regular ones. (5) Comparing Row 5 and 6, the supervised gIoU loss boosts the realism and accuracy by a significant margin, yet seemingly contradicts the regularity. (6) Fortunately, in Row 7, adding overlap loss and misalignment loss optimizes all the metrics. We name this "LayoutDETR-GAN (ours)" and stick to it for the following experiments. (7) Error margins after "$\pm$" are consistently smaller than value differences across rows, indicating the differences are statistically significant.

**Conditional embedding configurations.** We do not consider foreground images in this ablation study as the embeddings are simply image features. Whereas for foreground texts, we examine the importance of text length embeddings and text class embeddings by progressively removing them from the training. From Table 2 bottom section we observe: (1) Comparing Row 7 and 8, text length embeddings are beneficial all around. We reason that text length is a strong indicator of a proper text box size. This validates the strong positive correlation between text lengths and box sizes. (2) Comparing Row 8 and 9, text label embeddings serve as an essential role in the generation. Layout FID and Layout KID deteriorate significantly without text label embeddings (Row 9) as bounding boxes of similar texts tend to collapse to the same regions (referring to the 0.0 misalignment). This implies that text content itself is not as discriminative as text labels to differentiate box parameters.

Table 3: Quantitative comparisons to baselines. Each cell contains mean±std. For FIDs and KIDs, they are the statistics over 10 runs. ⇑/⇓ indicates a higher/lower value is better. **Bold**/underline font indicates the top/second best value in the column.

| Method | Realism | | | | Accuracy | | Regularity | |
|---|---|---|---|---|---|---|---|---|
| | Layout FID ⇓ | Layout KID $(\times 10^{-3})$⇓ | Image FID ⇓ | Image KID $(\times 10^{-5})$⇓ | IoU ⇑ | DocSim ⇑ | Overlap ⇓ | Misalign $(\times 10^{-2})$⇓ |
| **Our ad banner dataset** | | | | | | | | |
| LayoutGAN++ | $4.25_{\pm 0.01}$ | $16.62_{\pm 0.05}$ | $28.40_{\pm 0.06}$ | $58.54_{\pm 1.45}$ | $0.163_{\pm 0.002}$ | $0.130_{\pm 0.001}$ | $0.104_{\pm 0.003}$ | $0.759_{\pm 0.021}$ |
| READ | $4.45_{\pm 0.02}$ | $15.21_{\pm 0.21}$ | $32.10_{\pm 0.13}$ | $77.53_{\pm 2.23}$ | $0.177_{\pm 0.002}$ | $0.141_{\pm 0.001}$ | **$0.093_{\pm 0.002}$** | $2.867_{\pm 0.040}$ |
| Vinci | $38.97_{\pm 0.10}$ | $231.70_{\pm 1.22}$ | $58.12_{\pm 0.20}$ | $833.00_{\pm 3.55}$ | $0.104_{\pm 0.001}$ | $0.143_{\pm 0.001}$ | $0.243_{\pm 0.003}$ | **$0.271_{\pm 0.010}$** |
| LayoutTransformer | $5.47_{\pm 0.01}$ | $13.87_{\pm 0.01}$ | $39.70_{\pm 0.01}$ | $134.87_{\pm 1.03}$ | $0.080_{\pm 0.001}$ | $0.115_{\pm 0.001}$ | $0.127_{\pm 0.003}$ | $3.632_{\pm 0.065}$ |
| CGL-GAN | $4.69_{\pm 0.01}$ | $17.58_{\pm 0.02}$ | $30.50_{\pm 0.02}$ | $13.52_{\pm 1.40}$ | $0.154_{\pm 0.002}$ | $0.127_{\pm 0.001}$ | $0.116_{\pm 0.003}$ | $1.191_{\pm 0.025}$ |
| ICVT | $12.54_{\pm 0.06}$ | $64.49_{\pm 0.12}$ | $30.11_{\pm 0.05}$ | $62.29_{\pm 2.54}$ | $0.163_{\pm 0.002}$ | $0.137_{\pm 0.001}$ | $0.423_{\pm 0.006}$ | $0.682_{\pm 0.018}$ |
| LayoutDETR-GAN | **$3.19_{\pm 0.01}$** | **$5.62_{\pm 0.01}$** | **$27.35_{\pm 0.04}$** | $8.31_{\pm 0.80}$ | $0.151_{\pm 0.000}$ | $0.151_{\pm 0.000}$ | $\underline{0.101}_{\pm 0.003}$ | $\underline{0.646}_{\pm 0.011}$ |
| LayoutDETR-VAE | $3.25_{\pm 0.03}$ | $11.97_{\pm 0.26}$ | $\underline{27.47}_{\pm 0.04}$ | $\underline{7.70}_{\pm 0.22}$ | **$0.216_{\pm 0.002}$** | **$0.152_{\pm 0.001}$** | $0.119_{\pm 0.002}$ | $1.737_{\pm 0.037}$ |
| LayoutDETR-VAE-GAN | $\underline{3.23}_{\pm 0.02}$ | $\underline{10.75}_{\pm 0.09}$ | $27.88_{\pm 0.11}$ | **$4.18_{\pm 0.24}$** | $\underline{0.210}_{\pm 0.002}$ | $\underline{0.151}_{\pm 0.001}$ | $0.117_{\pm 0.002}$ | $1.439_{\pm 0.037}$ |
| **CGL Chinese ad banner dataset** | | | | | | | | |
| LayoutGAN++ | $11.43_{\pm 0.05}$ | $59.02_{\pm 0.26}$ | $11.92_{\pm 0.05}$ | $1082.68_{\pm 20.71}$ | $0.061_{\pm 0.000}$ | $0.083_{\pm 0.000}$ | $0.593_{\pm 0.007}$ | $0.729_{\pm 0.017}$ |
| READ | $10.51_{\pm 0.04}$ | $107.30_{\pm 1.24}$ | $6.58_{\pm 0.08}$ | $465.96_{\pm 14.92}$ | **$0.269_{\pm 0.002}$** | **$0.127_{\pm 0.001}$** | $0.145_{\pm 0.002}$ | $0.704_{\pm 0.098}$ |
| Vinci | $12.06_{\pm 0.01}$ | $80.45_{\pm 0.43}$ | $5.38_{\pm 0.02}$ | $320.99_{\pm 6.30}$ | $\underline{0.266}_{\pm 0.002}$ | $\underline{0.125}_{\pm 0.001}$ | **$0.093_{\pm 0.002}$** | $0.433_{\pm 0.042}$ |
| LayoutTransformer | $5.11_{\pm 0.01}$ | $33.72_{\pm 0.54}$ | $4.65_{\pm 0.01}$ | $286.08_{\pm 2.99}$ | $0.186_{\pm 0.002}$ | $0.114_{\pm 0.001}$ | $0.340_{\pm 0.005}$ | **$0.276_{\pm 0.027}$** |
| CGL-GAN | $5.63_{\pm 0.01}$ | $36.99_{\pm 0.30}$ | $7.26_{\pm 0.09}$ | $744.49_{\pm 12.73}$ | $0.107_{\pm 0.001}$ | $0.093_{\pm 0.001}$ | $0.297_{\pm 0.004}$ | $0.538_{\pm 0.011}$ |
| ICVT | $10.76_{\pm 0.14}$ | $119.10_{\pm 1.14}$ | $4.22_{\pm 0.05}$ | $109.33_{\pm 3.85}$ | $0.169_{\pm 0.002}$ | $0.109_{\pm 0.001}$ | $0.327_{\pm 0.004}$ | $\underline{0.340}_{\pm 0.060}$ |
| LayoutDETR-GAN | **$2.40_{\pm 0.01}$** | $\underline{13.60}_{\pm 0.29}$ | $\underline{4.11}_{\pm 0.06}$ | $\underline{22.60}_{\pm 0.79}$ | $0.157_{\pm 0.002}$ | $0.106_{\pm 0.000}$ | $0.187_{\pm 0.003}$ | $0.464_{\pm 0.010}$ |
| LayoutDETR-VAE | $8.57_{\pm 0.10}$ | $94.84_{\pm 1.09}$ | $4.21_{\pm 0.03}$ | $144.27_{\pm 5.18}$ | $0.208_{\pm 0.002}$ | $0.120_{\pm 0.001}$ | $0.288_{\pm 0.003}$ | $0.374_{\pm 0.064}$ |
| LayoutDETR-VAE-GAN | $\underline{2.65}_{\pm 0.05}$ | **$12.37_{\pm 0.60}$** | **$2.66_{\pm 0.02}$** | **$19.30_{\pm 1.04}$** | $0.180_{\pm 0.002}$ | $0.110_{\pm 0.001}$ | $\underline{0.134}_{\pm 0.002}$ | $0.401_{\pm 0.070}$ |
| **CLAY mobile application UI dataset** | | | | | | | | |
| LayoutGAN++ | $14.12_{\pm 0.06}$ | $60.20_{\pm 0.52}$ | $7.49_{\pm 0.02}$ | $148.33_{\pm 4.02}$ | $0.049_{\pm 0.001}$ | $0.078_{\pm 0.001}$ | $0.817_{\pm 0.115}$ | $1.057_{\pm 0.028}$ |
| READ | $3.68_{\pm 0.01}$ | $21.98_{\pm 0.15}$ | $5.38_{\pm 0.02}$ | $91.43_{\pm 3.24}$ | $\underline{0.312}_{\pm 0.001}$ | $\underline{0.121}_{\pm 0.001}$ | $\underline{0.099}_{\pm 0.002}$ | $2.045_{\pm 0.045}$ |
| Vinci | $22.98_{\pm 0.02}$ | $216.90_{\pm 0.66}$ | $13.04_{\pm 0.05}$ | $677.42_{\pm 7.22}$ | $0.178_{\pm 0.002}$ | $0.104_{\pm 0.001}$ | $0.253_{\pm 0.004}$ | $2.526_{\pm 0.055}$ |
| LayoutTransformer | $\underline{2.64}_{\pm 0.01}$ | $\underline{5.03}_{\pm 0.18}$ | $5.27_{\pm 0.01}$ | $\underline{55.99}_{\pm 2.49}$ | $0.216_{\pm 0.003}$ | $0.106_{\pm 0.002}$ | $0.357_{\pm 0.006}$ | $0.833_{\pm 0.032}$ |
| CGL-GAN | $47.74_{\pm 0.02}$ | $190.96_{\pm 1.11}$ | $8.96_{\pm 0.02}$ | $226.81_{\pm 4.65}$ | $0.034_{\pm 0.001}$ | $0.066_{\pm 0.000}$ | $1.153_{\pm 0.141}$ | $1.099_{\pm 0.011}$ |
| ICVT | $4.56_{\pm 0.04}$ | $18.35_{\pm 0.26}$ | $\underline{5.26}_{\pm 0.03}$ | $69.83_{\pm 2.24}$ | $0.208_{\pm 0.002}$ | $0.105_{\pm 0.001}$ | $0.396_{\pm 0.006}$ | $1.066_{\pm 0.043}$ |
| LayoutDETR-GAN | **$1.84_{\pm 0.02}$** | **$3.01_{\pm 0.15}$** | $5.22_{\pm 0.02}$ | **$11.19_{\pm 1.39}$** | $0.261_{\pm 0.003}$ | $0.113_{\pm 0.001}$ | **$0.083_{\pm 0.002}$** | $\underline{0.773}_{\pm 0.016}$ |
| LayoutDETR-VAE | $4.99_{\pm 0.01}$ | $30.18_{\pm 0.32}$ | $5.49_{\pm 0.03}$ | $107.55_{\pm 3.57}$ | **$0.327_{\pm 0.003}$** | **$0.123_{\pm 0.001}$** | $0.205_{\pm 0.004}$ | $5.119_{\pm 0.019}$ |
| LayoutDETR-VAE-GAN | $3.98_{\pm 0.12}$ | $18.39_{\pm 0.98}$ | $5.87_{\pm 0.02}$ | $82.31_{\pm 1.56}$ | $0.158_{\pm 0.002}$ | $0.108_{\pm 0.001}$ | $0.148_{\pm 0.002}$ | **$0.691_{\pm 0.030}$** |

## 5.2 COMPARISONS TO BASELINES

**Quantitative** comparisons are in Table 3. We evaluate three of our alternative model variants: GAN-, VAE-, and VAE-GAN-based. We observe: (1) On our ad banner dataset, at least one of our variants outperforms all the baselines in terms of realism and accuracy. Our LayoutDETR-GAN achieves the second best in terms of regularity. The margin to the best baseline is minor. This evidences the efficacy of our understanding of multi-modality and sophisticated loss configurations. (2) On CGL Chinese ad banner dataset, our variants lead in the most important realism metrics by significant margins. Our LayoutDETR-VAE-GAN variant is the best overall and outperforms the LayoutDETR-GAN variant. This is because CGL has more elements per sample and VAE plays a more important role than GAN for complex layout arrangements. LayoutTransformer achieves a similarly balanced performance. READ and Vinci lead in the accuracy metrics yet significantly underperform in at least one other metric. (3) On CLAY dataset, at least one of our variants outperforms all the baselines in all metrics. Our efficacy generalizes in at least these two multimodal foreground domains: texts and images. (4) Error margins after "$\pm$" are consistently smaller than value differences across rows, indicating the differences are statistically significant.

**Qualitative** comparisons are in Fig. 3. We observe: (1) For our and CGL ad banner datasets, our designs understand the background the most effectively. For example, they never overlay foreground elements on top of clutter background subregions. If the background looks symmetric, our layouts are placed in the middle of the banners. (2) For these two datasets, our designs also approximate the real layout distribution the most closely, in terms of the relative font sizes (box sizes) and spatial relations (box orders and distances). Even for samples not close to their ground truth, our design variants still look the most reasonable and most harmonic together with backgrounds. (3) CLAY dataset is more challenging in terms of multimodal conditioning, as it has more tiny foreground elements. Still, our layouts appear the best aligned and least overlapping, with reasonable designs to harmonize with backgrounds, although different from the ground truth. Appendix Fig. 7-9 and the attached video show more uncurated results, including the impact of varying texts on layouts and our limitation in challenging scenarios.

## 5.3 GRAPHICAL SYSTEM DESIGN AND USER STUDY

We integrate generators into a graphical system for user-friendly applications in practice. The UI design is in Section D in Appendix and the demo video is attached in the supplementary material. With the graphical system, we can test a massive number of cases and collect generation results in the wild. This facilitates us to analyze users' subjective preferences for our designs. In specific, we tested on 308 ad banner samples by rendering our designs and baseline designs via the graphical system. We configured a binary comparison task between each pair of designs. In total, we launched

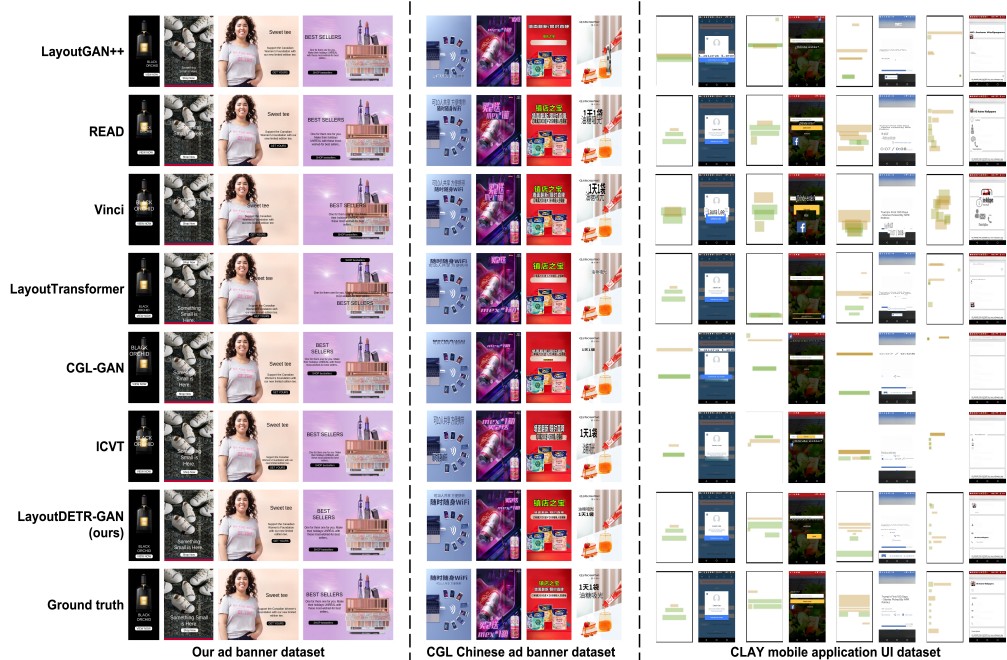

Figure 3: **Left**: comparisons on the testing set of **our ad banner dataset**. We apply the same rendering process to all methods: (1) Text font sizes and line breakers are adaptively determined to tightly fit into their inferred boxes. (2) Text font colors and button pad colors are adaptively determined to be either black or white whichever contrasts more with the background. (3) Button text colors are then determined to contrast with the button pads. (4) Text font is set to *Arial*. (5) Boxes are enforced to horizontally center-align with each other. **Middle**: comparisons on **CGL Chinese ad banner dataset**. Image patches that contain foreground text elements are resized and overlaid on the background following the generated layouts. **Right**: comparisons on **CLAY mobile application UI dataset**.

Table 4: Pairwise user preferences (column method over row method) on our ad banner dataset.

| Method | READ | Vinci | LayoutTransformer | CGL-GAN | ICVT | LayoutDETR-GAN (ours) |
|---|---|---|---|---|---|---|
| LayoutGAN++ | $49.8\%_{p=0.4}$ | $45.6\%_{p=3e-3}$ | $44.4\%_{p=3e-4}$ | $53.9\%_{p=0.01}$ | $47.1\%_{p=0.04}$ | $55.7\%_{p=2e-4}$ |
| READ | – | $45.1\%_{p=1e-3}$ | $44.5\%_{p=3e-4}$ | $53.8\%_{p=0.01}$ | $53.0\%_{p=0.04}$ | $54.2\%_{p=5e-3}$ |
| Vinci | – | – | $51.7\%_{p=0.2}$ | $55.8\%_{p=2e-4}$ | $56.9\%_{p=1e-5}$ | $62.6\%_{p=3e-15}$ |
| LayoutTransformer | – | – | – | $57.1\%_{p=8e-6}$ | $56.0\%_{p=2e-4}$ | $63.5\%_{p=2e-17}$ |
| CGL-GAN | – | – | – | – | $48.9\%_{p=0.2}$ | $54.7\%_{p=3e-3}$ |
| ICVT | – | – | – | – | – | $55.4\%_{p=6e-4}$ |

$308 \times \binom{7}{2} = 6,468$ jobs on AMT, and randomly assigned three workers for each job. We simply asked each worker "Which of the two images looks better?" We intentionally did not pre-define any criterion for the preference so as to fully respect their subjective judgments. All workers have an approval rate history above 90% on AMT. We did not set any restrictions for workers' gender, race, sexuality, demographics, locations, remuneration rates, etc. Our user study has been reviewed and approved by our ethical board.

Table 4 lists the ratio of users that prefer one design over another. To quantify the statistical significance of our user study, we calculate the p-value of a null hypothesis that the results of binary comparisons are equivalent to tossing a fair coin. We calculate it via the cumulative distribution function of binomial distribution, the smaller the more significant. In the last column all ratios are above 50%: a majority of users prefer our designs over any baseline significantly, considering all p-values $\ll 0.05$. This validates that our layout designs are more appealing to users. We attribute this to our effective approximation of real layout distributions and multimodal understanding.

# 6 CONCLUSION

We present LayoutDETR for customizable layout design. It inherits the high quality and realism of generative models, and benefits from object detectors to understand multimodal conditions. Experiments show that we achieve a new state-of-the-art performance for layout generation on a new ad banner dataset and beyond. We implement our solution as a graphical system that facilitates user studies. Users prefer our designs over several recent works. **Future work** includes: (1) exploring diffusion models (Song et al., 2021; Rombach et al., 2022) for multimodal layout generation; (2) establishing datasets that benchmark multimodal layout generation for 2D/3D natural scenes.

## REPRODUCIBILITY STATEMENT

In accordance with the principles of open science and with the aim of promoting reproducibility, transparency, and follow-up research, we commit to granting open-source access to all the materials associated with our study. Part of our dedication is underscored in Appendix, including the implementation details in Section B (where we integrate the following GitHub repositories: StyleGAN3 (Sty), LayoutGAN++ (Lay), DETR (DET), UP-DETR (UP-), BLIP (BLI)), dataset collection details in Section C, and graphical system design details in Section D. Our code and demo video of the system have been attached in the supplementary material. We promise to release our dataset, well-trained models, and fully-documented system. In addition to offering these resources, we commit to their maintenance and providing necessary support. Our goal is to contribute to our research community, making it more supportive and inclusive.

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

## A  Motivations of Using Each Network Component

The generator $G$ and conditional/unconditional discriminators $D^c/D^u$ serve as the fundamental components for the GAN variant of our solution. The use of conditional discriminator $D^c$ is straightforward due to the nature of our multimodal conditions: it encourages foreground elements to be harmonic and reasonable to the background after being overlaid according to the generated layout. The unconditional discriminator $D^u$ differs from $D^c$ by ignoring background and foreground elements, and focusing only on the realism of layouts themselves. Therefore, the use of $D^u$ additionally encourages the mutual relations among bounding boxes in a generated layout to be realistic and reasonable, regardless of multimodal conditions. The empirical effectiveness of $D^u$ is evidenced in Table 2 Row 5 in the main paper.

The motivation of using auxiliary decoders $F^c/F^u$ following $D^c/D^u$ is inspired by (Kikuchi et al., 2021). $F^c/F^u$ targets to self-reconstruct all the input information of $D^c/D^u$ through the bottleneck discriminator features. It encourages the input information to be fully encoded into the discriminator features such that the discriminator classification is fully justified by the input. The empirical effectiveness of $F^c/F^u$ has been validated in (Kikuchi et al., 2021) Table 2 last column.

Positional embeddings $\mathcal{E}^c/\mathcal{E}^u$ in the auxiliary decoders $F^c/F^u$ are inherited from LayoutGAN++ (Kikuchi et al., 2021) and are necessary, without which the reconstructed bounding boxes in a layout would have no variance as they are conditioned on the same discriminator's final features $\mathbf{f}^c/\mathbf{f}^u$. They differ from the positional embeddings in the visual transformer (ViT) background encoder. The latter is used to differentiate patch embeddings in different image coordinates.

The layout encoder $E$ and generator $G$ serve as the fundamental components for the VAE variant of our solution. VAEs and GANs are alternative paradigms of generative models, and are complementary to each other in terms of data distribution learning and feature representation learning (Larsen et al., 2016). Inspired by (Zheng et al., 2019), we combine the strengths of VAEs and GANs and apply them for layout generation.

The use of auxiliary reconstructor $R$ following $G$ stems from the same motivation as the use of auxiliary decoders $F^c/F^u$. Its empirical effectiveness is evidenced in Table 2 Row 4 in the main paper.

The motivation of using DETR architecture (Carion et al., 2020) for background image encoding and understanding stems from its state-of-the-art performance for object detection using the state-of-the-art ViT encoder architecture (Wang et al., 2018; He et al., 2022). In our scenarios, "detection" is equivalent to "generation" as both processes output bounding box parameters. Although "detection" lacks the "layout" concept, it is complemented by the layout discriminator and encoder networks. "Object" stands for the non-clutter subregions in a background that are suitable for overlaying foreground elements. ViT tokenizes visual features that facilitate cross attention with other foreground element features, e.g., tokenized text features, so that multimodal conditions can synergize jointly.

Incorporating DETR into multimodal layout generation is non-trivial. Directly applying the DETR architecture in the encoder did not achieve optimal performance. Tuning around different generator paradigms (Sec. 3.1), loss configurations (Sec. 5.2), and conditional embedding configurations (Sec. 5.2) leads us towards the empirical optimum.

## B  Implementation Details

Architecture design is where we integrate object detection with layout generation. Detection transformer (DETR) architecture (Carion et al., 2020) is employed and modified for LayoutDETR generator $G$ and conditional discriminator $D^c$. It targets to boost the understanding of background from the perspective of visual detection, and enhance the controllability of background on the layout.

As depicted in Figure 2 bottom left in the main paper, $G$ and $D^c$ contain a visual transformer encoder for background understanding and a transformer decoder for layout generation or discriminator feature representation. The encoder part is the same as in DETR (Carion et al., 2020), and is identical in $G$ and $D^c$. It consists of a CNN backbone that extracts a compact feature representation from a background image, as well as a multi-head ViT encoder (Wang et al., 2018; He et al., 2022) that incorporates positional encoding inputs (Parmar et al., 2018; Bello et al., 2019). It outputs tokenized visual features for cross-attention in the following layout transformer decoder.

The layout decoder is also inherited from the DETR transformer decoder with self-attention and encoder-decoder-cross-attention mechanisms (Vaswani et al., 2017). In $G$, it transforms each of the $N$ input embeddings (corresponding to $N$ foreground elements) into layout bounding box parameters, whereas in $D^c$, it transforms each of the $N$ bounding box embeddings into discriminator features. Our architecture differs from DETR where we have foreground elements as inputs to drive the transformation, while DETR does not. Therefore, we replace their freely-learnable object queries with our foreground embeddings as the input tokens to the decoder, which are detailed below.

In $G$, foreground elements are composed of texts $\mathcal{T} = \{\mathbf{t}^i\}_{i=1}^M = \{(\mathbf{s}^i, c^i, l^i)\}_{i=1}^M$ and image patches $\mathcal{P} = \{\mathbf{p}^i\}_{i=1}^K$. Thence, each foreground embedding is a concatenation of noise embedding and either text embedding or image embedding. To calculate the text embedding, we separately encode text string $\mathbf{s}$, text class $c$, and text length $l$, and concatenate the features together. The text string is encoded by the pretrained and fixed BERT text encoder (Devlin et al., 2019). The text class and quantized text length are encoded by learning a dictionary. To calculate the image embedding, we use the same ViT as used for background encoding. The weights are shared and initialized by the Up-DETR-pretrained model (Dai et al., 2021). Note that the font color is not considered in the modeling because it is trivial information. According to our empirical observation, font colors are dominated by two and only two modes: black and white. As indicated in Fig. 3 caption in the main paper: Text font colors and button pad colors are adaptively determined to be either black or white whichever contrasts more with the background.

For the other networks $F^c$, $D^u$, $F^u$, $E$, and $R$ that do not take background images as an input condition, we simply use the above transformer decoder architecture as their implementations. Following transformers, for each foreground image reconstruction in $F^c$ and $R$, we employ the StyleGAN2 image generator architecture (Karras et al., 2020). For each text string reconstruction, we employ the pretrained BERT language model decoder (Devlin et al., 2019; Li et al., 2022b). For each text class and text length decoding, we use 3-layer MLPs.

We implement LayoutDETR in PyTorch and use Adam optimizer (Kingma & Ba, 2015) to train the models on 8 NVIDIA A100 GPUs, each with 40GB memory. Because bounding box parameters are normalized by image resolutions, during training and inference we downsize arbitrary images to $256 \times 256$ without losing generality. For final rendering and visualization, we resize them back to their original resolutions. Small and unified image size allows us to train models with a large enough batch size, e.g., 64 in our experiments. During training, we set the learning rate constantly as $10^{-5}$ and train for 110k iterations in 4 days. Inference is much more efficient: we load only $G$ into a single NVIDIA A100 GPU and it consumes only 2.82GB of memory. It takes only 0.38 sec to generate a layout given foreground and background conditions.

The training and inference code is attached with the supplementary material.

## C  DETAILS OF OUR AD BANNER DATASET COLLECTION

The sources of raw ad banner images consist of two parts. First, we manually went through all the images in Pitt Image Ads Dataset (Hussain et al., 2017). We filtered out those with single modality, low quality, or old-fashioned designs. We then selected 3,536 valid ad banner images. Second, we searched on Google Image Search Engine with the keywords "XXX ad banner" where "XXX" goes through a list of 2,765 retailer brand names including the Fortune 500 brands. For each keyword search, we crawled the top 20 results and manually filtered out non-ads, single-modality, low-quality, or offensive-content images. We then selected 4,321 valid ad banner images. Combining the two sources, we in total obtained 7,857 valid ad banner images with arbitrary sizes.

Next, we crowdsourced on Amazon Mechanical Turk (AMT) (mtu) to obtain human annotations for the bounding box and class of each text phrase in each image. The class space spans over 11 categories as shown in the AMT interface in Figure 4 top. Without losing representativeness, we focus on the top-4 most common categories in this work: {*header*, *body text*, *disclaimer / footnote*, *button*}. We also linked a detailed instructional document with examples for workers to fully understand the annotation task. See the instruction in Figure 4 bottom. We assigned each annotation job to three workers. For the final annotation results, we averaged over three workers' submissions and incorporated our judgments for the tie cases. In total there were 67 workers involved in the task. On average each worker submitted 314 jobs in around 3 minutes per job. All of the workers have an approval rate history above 90% on AMT. We did not set any restrictions for workers' gender,

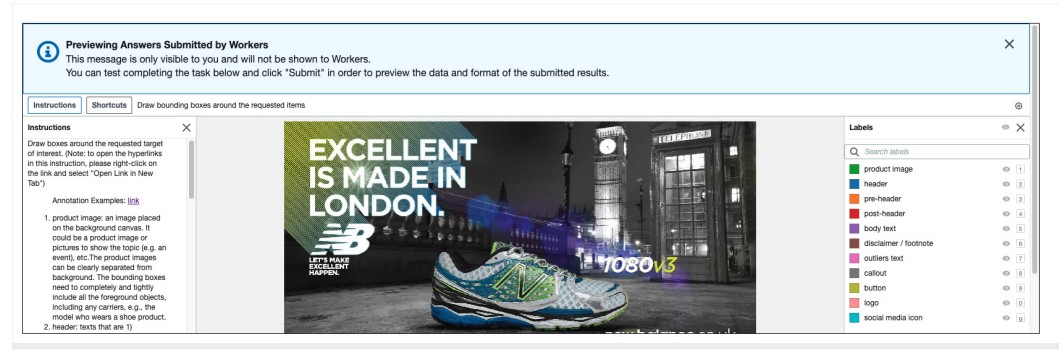

(a) AMT interface

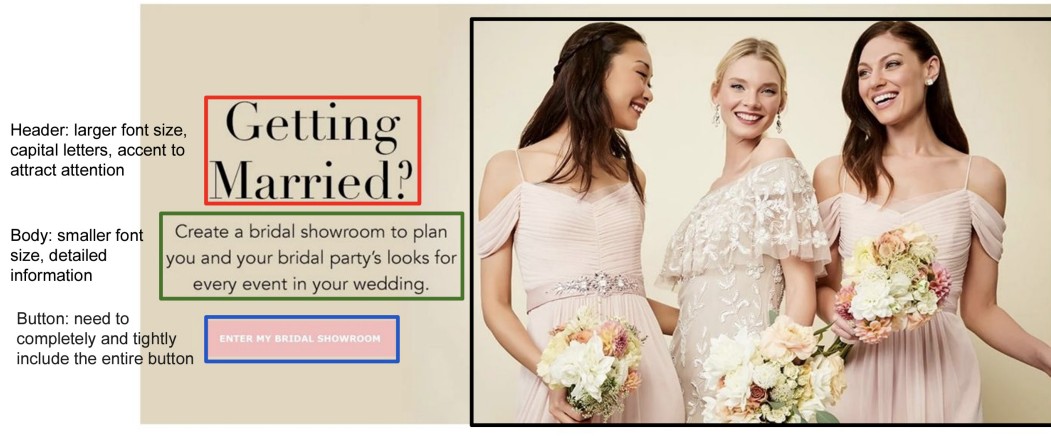

(b) Instructional example

Figure 4: **Top**: AMT interface with instructions on the left for users to annotate the bounding box and class of each existing copywriting text on each image. **Bottom**: one instructional example of the definitions of text bounding boxes and text classes.

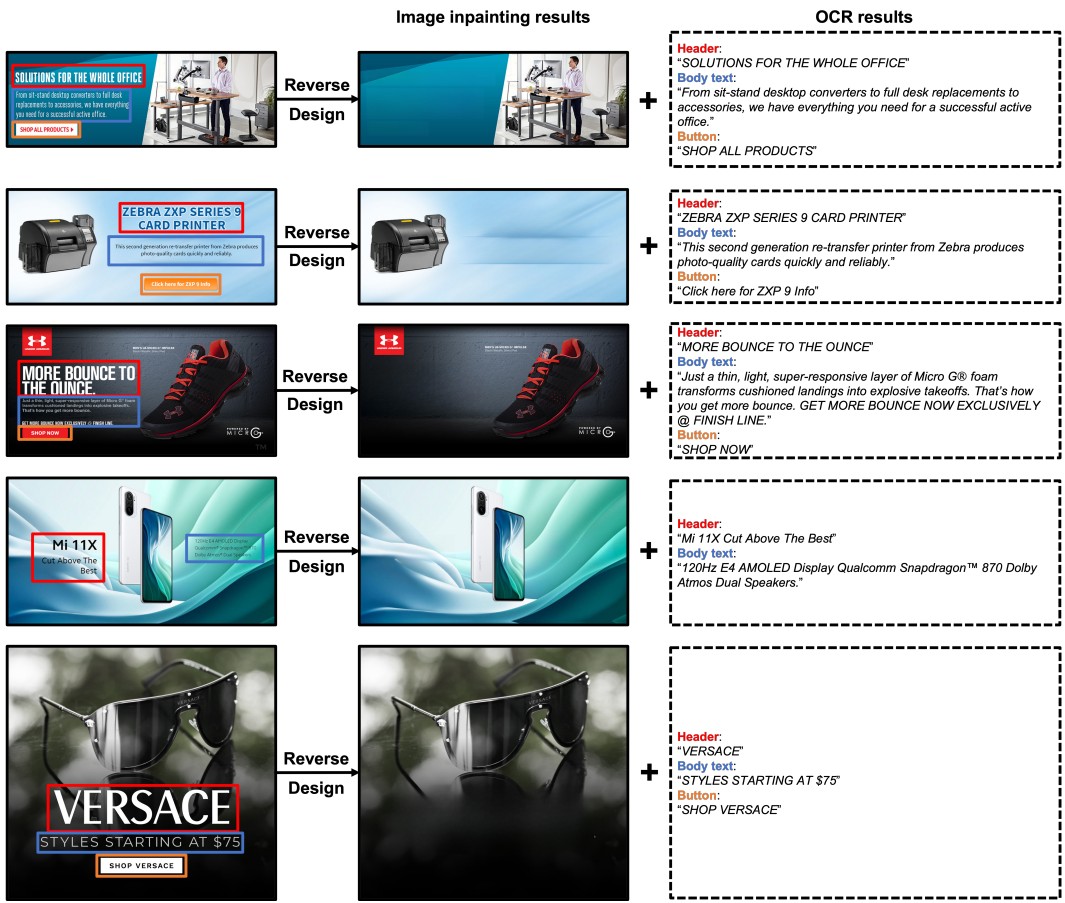

Figure 5: Reverse engineering examples of separating foreground elements from background images using OCR and image inpainting techniques.

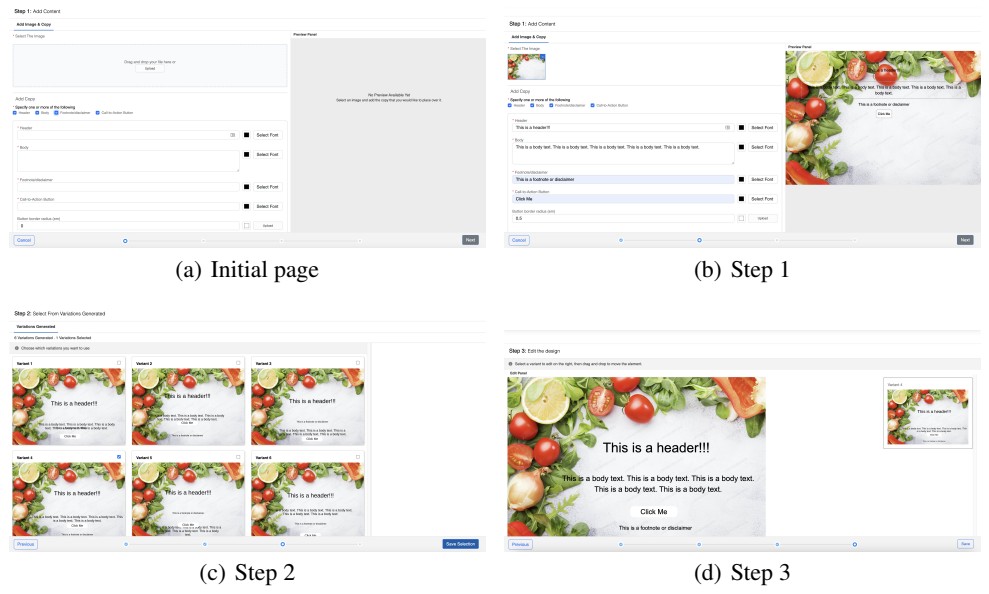

(a) Initial page

(b) Step 1

(c) Step 2

(d) Step 3

Figure 6: Step-by-step usage of our graphical system for customizable multimodal graphic layout design.

race, sexuality, demographics, locations, remuneration rates, etc. Our annotation process has been reviewed and approved by our ethical board.

After annotation, it is necessary to reverse the design by separating foreground elements from background images to configure the training/testing data. We apply a modern optical character recognition (OCR) technique (pad) to extract the text inside each bounding box, and adopt a modern image inpainting technique (Suvorov et al., 2022) to erase the texts. The separation of texts from background images is exemplified in Figure 5. After filtering out a few samples with undesirable OCR or inpainting results, we finally obtain 7,196 valid samples for the following experiments.

It is worth noting that inpainting clues may leak the layout bounding box ground truth information and shortcut training. Therefore, during training, we intentionally inpaint background images at additional random subregions that are irrelevant to their layouts.

## D GRAPHICAL SYSTEM STEP-BY-STEP DESIGNS

Since we validate that our solution sets up a new state of the art for multimodal layout design, it is worth integrating it into a graphical system for user-friendly service in practice. Figure 6 demonstrates our step-by-step UI designs. In specific:

(1) Figure 6(a) shows the initial page that allows users to customize their background images and optionally foreground elements: *header text*, *body text*, *footnote/disclaimer text*, *button text*, as well as *button border radius* (zero radius means a rectangular button). Text colors, button pad colors, and text fonts can also be customized.

(2) Once users upload their background and foreground elements, they are previewed on the right part of the same page, as shown in Figure 6(b). The locations and sizes of foreground elements in the preview are meaningless: they just conceptually show what contents are going to be rendered on top of the background.

(3) Once users click "Next", it moves on to the next page with our design results, as shown in Figure 6(c). Given one layout designed in the backend, we post-process it by randomly jittering the generated box parameters by 20% while keeping the original non-overlapping and alignment regularity.

(4) Afterwards, the system renders foreground elements given the layout bounding boxes. Text font sizes and line breakers are adaptively determined so as to tightly squeeze into the boxes. Considering *header texts* usually have short strings yet are assigned with large boxes, their font sizes are naturally large enough. Text font styles can also randomly vary. This optional feature is shown in our supplementary video. We showcase six of our rendered results on this page, and allow users to select one or more satisfactory designs.

(5) Once users make their selection(s) and click "Save Selection", it moves onto the last page as shown in Figure 6(d). On this page, users are allowed to manually customize the size and location of each rendered foreground element. Once they finish, they click "Save" to exit our system.

More live demonstrations are nested in our supplementary video.

## E MORE QUALITATIVE RESULTS

We show in Fig. 7 more uncurated qualitative results of layout designs and text rendering on background images in the wild and on CGL Chinese ad banner inpainted background (Zhou et al., 2022). Conditioned on multiple text inputs in varying categories, our designs appear aesthetically appealing and harmonic between foreground and background.

We show in Fig. 8 the impact of varying texts on layouts given the same background image. We observe: (1) the scales of bounding boxes are adaptively proportional to the varying lengths of texts such that the font sizes remain approximately unchanged, and (2) the global and relative locations of bounding boxes are stable regardless of the changes of texts, which are reliably harmonic with the background structure.

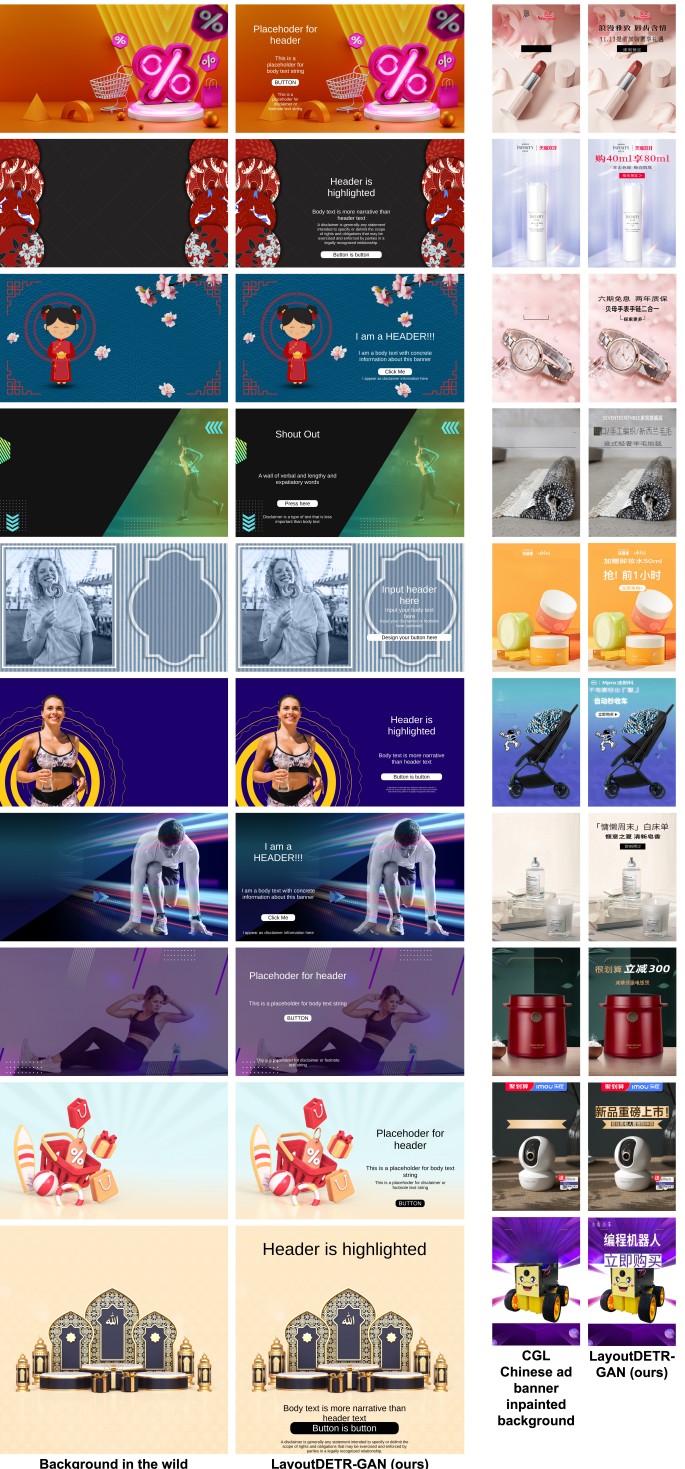

Figure 7: **Left**: uncurated layout designs and text rendering on background images in the wild (extracted from PSD data downloaded from (fre) with searching keywords "ad banner"). Rendering rules: (1) Text font sizes and line breakers are adaptively determined to tightly fit into their inferred boxes. (2) Text font colors and button pad colors are adaptively determined to be either black or white whichever contrasts more with the background. (3) Button text colors are then determined to contrast with the button pads. (4) Text font is set to *Arial*. **Right**: uncurated layout designs and text rendering on CGL Chinese ad banner background images inpainted by (Suvorov et al., 2022). Image patches that contain foreground text elements are resized and overlaid on the background following the generated layouts.

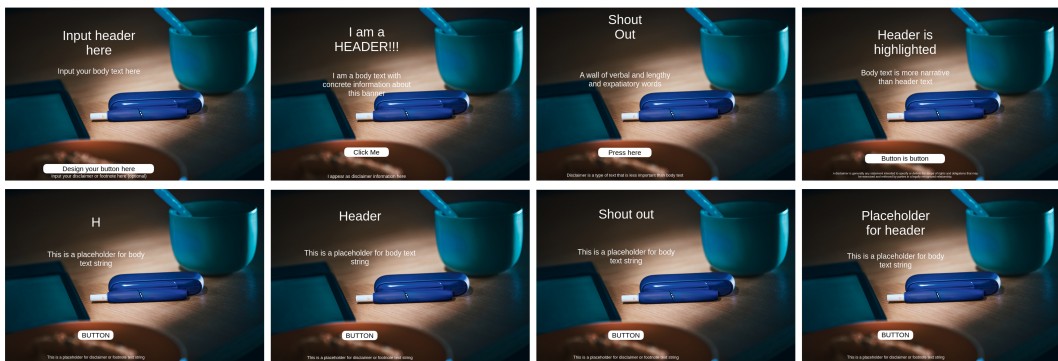

Figure 8: **Top**: ad banner design given the same background image (downloaded from (pmi)) and varying text combinations. **Bottom**: ad banner design given the same background image and varying only the *header* text component.

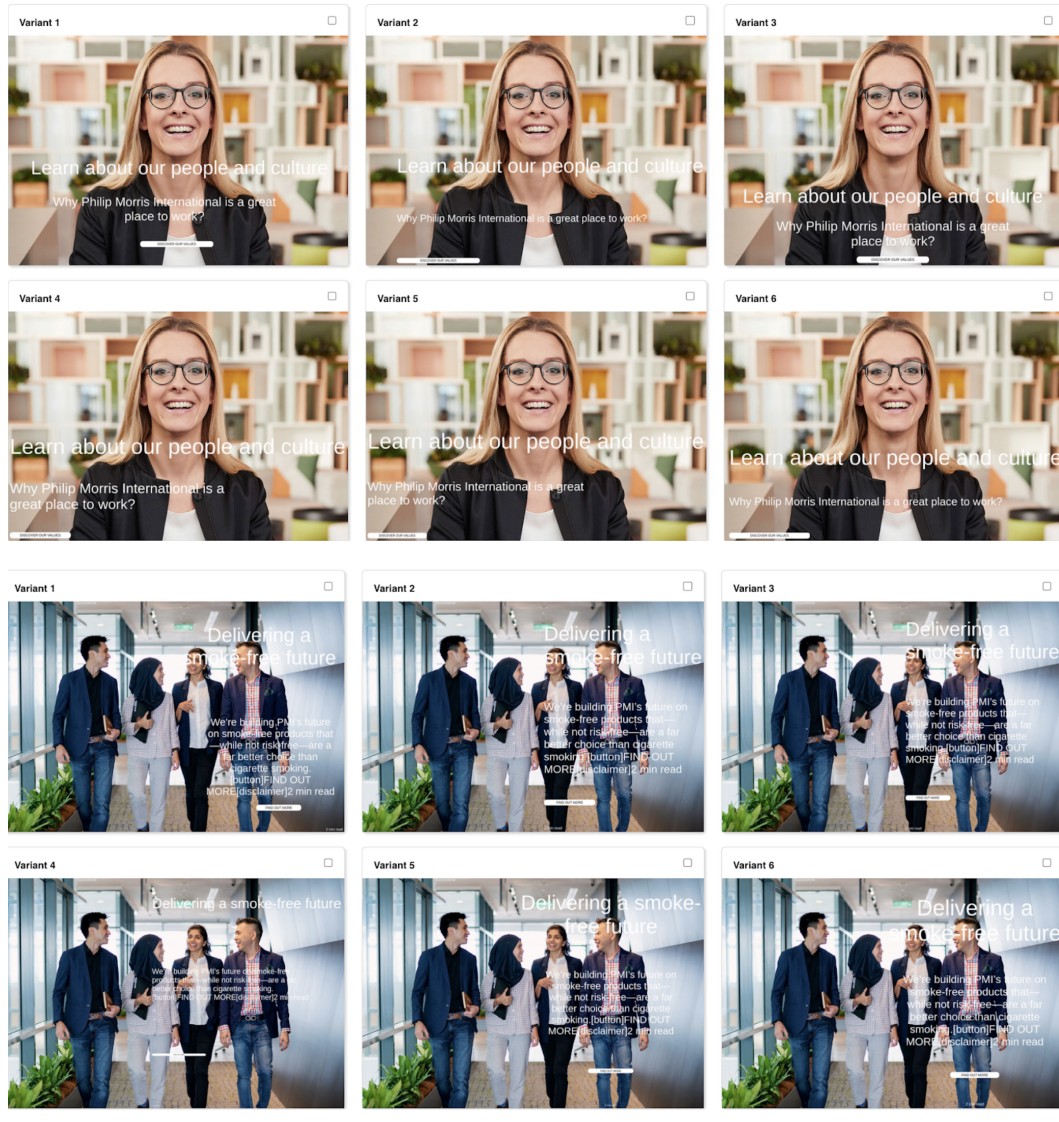

Figure 9: Imperfect layout designs for over-clutter background images and wordy texts. Image sources are from (pmi).

## F    LIMITATION ON CHALLENGING SAMPLES

We show in Fig. 9 a few imperfect layout designs for challenging samples. When the background images are over-clutter and texts are wordy, none of our rendering variants looks very ideal. Our model struggles between (1) placing layouts in the middle regardless of background and (2) placing layouts over less busy areas at edges that breaks the spatial balance. A possible workaround could be introducing gradient blending masks into the rendering post-processing.

