# OpenReview forum: "LayoutDETR: Detection Transformer Is a Good Multimodal Layout Designer"
_ICLR.cc/2024/Conference — Submitted to ICLR 2024_

### Official Review · Reviewer_t8xx · 2023-10-23

**Soundness:** 3 good
**Presentation:** 2 fair
**Contribution:** 2 fair
**Rating:** 5
**Confidence:** 4

**Summary:**

In this paper, VAE and GAN are combined with DETR to realize multimodal layout generation. A large-scale ad banner data set with 7,196 samples containing English characters is presented. According to the experimental results of three data sets on ad banner, CGL, and CLAY, the method achieves SOTA performance.

**Strengths:**

* The paper is easy to follow.

* A large-scale ad banner dataset is collected for the layout design task.

* The results show that the model achieves excellent performance.

**Weaknesses:**

* Many technical details are not well-motivated and validated, e.g., VAE and DETR structures.

* It seems the method combines multiple popular techniques and the novelty in the technical part is unclear.

* Simply considering the box layout and ignoring font information and box aspect ratios makes the task less extensible.

* The method requires dozens of loss functions for supervision. I am not sure how to tune weighting factors and make sure each term properly works.

**Questions:**

The importance and necessity of VAE design is not validated. As the method takes a generative pipeline, I am interested in the variations and the latent spaces. Moreover, a proper validation of this key design is also important.

---

> ### Author Response · Authors · 2023-11-16
> **Response to Reviewer t8xx**
>
> We thank all the reviewers for their constructive suggestions, which help improve the completeness of our submission. We are encouraged that the reviews are positive in the following five levels:
>
> * The paper is "**easy to follow**" (Reviewer t8xx).
> * The problem we are researching is “**important**” (Reviewer Gw2u) and our idea is “**interesting**” (Review j8Mq), “**effective**” (Review j8Mq), “**practically useful**” (Reviewer Gw2u), and is **one of our strengths** (Reviewer EPYk).
> * Our dataset contribution is “**useful**” (Reviewer j8Mq), “**valuable**” (Reviewer Gw2u),  and is **one of our strengths** (Reviewer EPYk, Reviewer t8xx).
> * Our evaluation is “**extensive**” (Reviewer Gw2u), and our results are “**SOTA**” (EPYk), “**effective**” (Reviewer j8Mq), “**excellent**” (Reviewer t8xx), and “**good**” (Reviewer Gw2u).
> * Our graphical system and user study are **one of the strengths** (Reviewer EPYk).
>
> We now address individual questions of **Reviewer t8xx** below.
>
> 1. **[Presentation: 2 fair?]**
>     - This **contradicts your first strength point** that “the paper is easy to follow”, and there is no specific weakness point or question about the presentation. We therefore appeal the reviewer’s re-evaluation about the “presentation” rating and final rating.
>
> 2. **[The VAE and DETR structures are not well-motivated and validated]**
>     - This is a **factual misunderstanding**. We have already discussed our motivations. Due to the space limit, we had to mention them in **the original submission Appendix Sec. A**. For VAE, our motivation has been articulated in **Paragraph 4**. VAE has been pre-validated by Larsen et al 2016 for image generation, and has been validated by us in **Table 3 LayoutDETR-VAE rows and LayoutDETR-VAEGAN rows**. With the help of VAE, the best results (in bold) are achieved in several metrics on several datasets. For DETR, our motivation has been articulated in **Paragraph 6 in the original submission Appendix Sec. A**. DETR has been pre-validated by Carion et al 2020 for object detection, and has been validated by CGL-GAN and ICVT for layout generation (see **Table 1**). We will manage to move this part to the main paper. We therefore appeal the reviewer’s re-evaluation about the “soundness” rating and final rating.
>
> 3. **[Technical novelty]**
>     - We understand this is a subjective judgement, and the debate would never end. We would rather argue using Prof. Michael Black’s Guide to Reviewers: Novelty in Science (https://medium.com/@black_51980/novelty-in-science-8f1fd1a0a143). We quote some:
>         - About the novelty of our connections between two established fields, multimodal conditioned layout generation and visual detection, Prof. Michael Black says “The novelty arose from the fact that nobody had put these ideas together before.” “Fortunately, these connections also turned out to be valuable, resulting in practical algorithms that were state of the art.” “To see the connections for the first time, before others saw them, was like breathing for the first time.” “The resulting paper embodies the translation of the idea into code, experiments, and text. In this translation, the beauty of the spark may be only dimly glimpsed. My request of reviewers is to try to imagine the darkness before the spark.”
>         - About the novelty of our simple and reasonable reuse of LayoutGAN++ and DETR architectures, Prof. Michael Black says “I value simplicity over unnecessary complexity; the simpler the better. Taking an existing network and replacing one thing is better science than concocting a whole new network just to make it look more complex.” “If a paper has a simple idea that works better than the state of the art, then it is most likely not trivial. The authors are onto something and the field will be interested.” “The inventive novelty was to have the idea in the first place. If it is easy to explain and obvious in hindsight, this in no way diminishes the creativity (and novelty) of the idea.”
>         - About the novelty of our newly-collected dataset, Prof. Michael Black says “Novelty (and value) come in many forms in papers. A new dataset can be novel if it does something no other dataset has done, even if all the methods used to generate the dataset are well known.”
>     - We therefore appeal the reviewer’s re-evaluation about the “contribution” rating and final rating.

---

> > ### Author Response · Authors · 2023-11-16
> > **Continued Response to Reviewer t8xx**
> >
> > 4. **[Simply considering the box layout and ignoring font information and box aspect ratios makes the task less extensible]**
> >     - This is a **factual misunderstanding**. Rather, we intend to learn the generator to implicitly automate the font size estimation and box aspect ratio estimation. Font size can be automatically determined by the generated box size and the length of the input texts. See **Fig. 3 caption (1) in the original submission**. Aspect ratio can be automatically determined by the generated box height and width. Our loss terms that are designed to approximate the real box parameters will implicitly learn reasonable font sizes and box aspect ratios during training and automatically determine them during inference. This has been validated by the qualitative results in **Fig. 3,7,8 and user study in Table 4 in the original submission**. All the font sizes and box aspect ratios look aesthetically pleasing. We therefore appeal the reviewer’s re-evaluation about the “soundness” rating and final rating.
> >
> > 5. **[The method requires dozens of loss functions. How to tune their weighting factors?]**
> >     - In fact, the effectiveness of loss terms have been ablation studied in **Table 2 Row 3-6 in the original submission**. Their hyper-parameter weighting factors have been discussed in **Sec. 3.2 last paragraph**: “All the $\lambda$s are trivially set to align the order of magnitude of each loss term. We use an identical set of $\lambda$s for all the datasets to validate our performance is insensitive to $\lambda$s.”
> >
> > 6. **[Variation of the generation]**
> >     - In fact, we have already discussed the variation of generation. Due to the space limit, we had to mention it in **the original submission Appendix Sec. E Paragraph 2 and Fig. 8**.

---

### Official Review · Reviewer_EPYk · 2023-10-29

**Soundness:** 2 fair
**Presentation:** 3 good
**Contribution:** 2 fair
**Rating:** 5
**Confidence:** 3

**Summary:**

This paper proposed LayoutDETR which can inherit high quality and realism from generative modeling, while reformulating content-aware requirements as a detection problem. It learns to detect in a background image the reasonable locations, scales, and spatial relations for multimodal foreground elements in a layout.

**Strengths:**

- study layout generation and visual detection with a unified framework
- proposed a new banner ads dataset
- achieve state of art in layout generation in terms of metrics of realism, accuracy, and regularity
- built graphical system and conduct user study

**Weaknesses:**

- the empty space detection on a background image and the layout generation of foreground can be decoupled as two separate steps. It is better to compare with such a baseline, and justify the superiority of doing it with a joint model.
- the proposed dataset (images) is collected in prior work. The new contribution here is the detected text objects, background inpainting and the text class annotation, which is not as significant as a new dataset.
- There are some concerns about the quality of data set. According to the way the data set was constructed, there are only texts as foreground objects, without other elements such as vector shape, image. This is very limited. Also, the inpainted background may contain artifacts which the generator can leverage for text location prediction. How is the text image patch obtained? If it's cropped from the original image, it has the same background patten, which may contain shortcut information for layout prediction.

**Questions:**

- please clarify whether there are only text as foreground object in the dataset and all the experiments.
- why Crello dataset is not multi modal? What is the unique part of the proposed data set?
- explain whether it's possible to apply this paper to the problem: "Towards Flexible Multi-modal Document Models"
- Eq 6, should it be p_1^i ?
- the paper does not evaluated diversity of the generated results. It would be good to show some visual examples of different design variations for one background image.

**Details Of Ethics Concerns:**

The banner ads images may contain copyright logos, faces, or other protected images.

---

> ### Author Response · Authors · 2023-11-16
> **Response to Reviewer EPYk**
>
> We thank all the reviewers for their constructive suggestions, which help improve the completeness of our submission. We are encouraged that the reviews are positive in the following five levels:
>
> * The paper is "**easy to follow**" (Reviewer t8xx).
> * The problem we are researching is “**important**” (Reviewer Gw2u) and our idea is “**interesting**” (Review j8Mq), “**effective**” (Review j8Mq), “**practically useful**” (Reviewer Gw2u), and is **one of our strengths** (Reviewer EPYk).
> * Our dataset contribution is “**useful**” (Reviewer j8Mq), “**valuable**” (Reviewer Gw2u),  and is **one of our strengths** (Reviewer EPYk, Reviewer t8xx).
> * Our evaluation is “**extensive**” (Reviewer Gw2u), and our results are “**SOTA**” (EPYk), “**effective**” (Reviewer j8Mq), “**excellent**” (Reviewer t8xx), and “**good**” (Reviewer Gw2u).
> * Our graphical system and user study are **one of the strengths** (Reviewer EPYk).
>
> We now address individual questions of **Reviewer EPYk** below.
>
> 1. **[Compare to the baseline: empty space detection on a background image and the layout generation of foreground can be decoupled as two separate steps]**
>     - In fact, we **have already compared to such a baseline and outperformed it (Table 1 in the original submission): CGL-GAN** does use saliency detection to extract the empty background first, and then learns the layout generation using GANs. We will highlight this in the next iteration. We therefore appeal the reviewer’s re-evaluation about the “soundness” rating and final rating.
>
> 2. **[The proposed dataset (images) is collected in prior work]**
>     - This is a **factual misunderstanding**. We did collect our own images. Due to the space limit, we had to mention it in **the original submission Appendix Sec. C Paragraph 1**: In the image level, we spent non-trivial manual efforts to go through all the images in Pitt Image Ads Dataset. We then thoughtfully filtered out those with single modality, low quality, or old-fashioned designs. We finally selected qualified 3,536 images out of noisy 64,832 images in the original dataset. Moreover, we additionally searched on Google Image Search Engine with the keywords ”XXX ad banner” where ”XXX” goes through a list of 2,765 retailer brand names including the Fortune 500 brands. For each keyword search, we crawled the top 20 results and manually filtered out non-ads, single-modality, low-quality, or offensive-content images. We then selected 4,321 valid ad banner images. Combining the two sources, we in total obtained 7,857 valid ad banner images with arbitrary sizes. We will manage to move this part to the main paper. We therefore appeal the reviewer’s re-evaluation about the “contribution” rating and final rating.
>
> 3. **[The inpainted background may contain artifacts which the generator can leverage for text location prediction]**
>     - This is a reasonable concern. In fact, we **have already considered and resolved it**. Due to the space limit, we had to discuss it in **the original submission Appendix C last paragraph**: “It is worth noting that inpainting clues may leak the layout bounding box ground truth information and shortcut training. Therefore, during training, we intentionally inpaint background images at additional random subregions that are irrelevant to their layouts.” As a result, such random training augmentations introduce additional possible inpainting artifacts that do not respond to layout locations. This avoids training from being overfitting to inpainting artifacts, and makes the inference not to focus on any such artifacts. We will move this part to the main paper. We therefore appeal the reviewer’s re-evaluation about the “soundness” rating and final rating.
>
> 4. **[The text patch has the same background patten, which may contain shortcut information for layout prediction]**
>     - This is a **factual misunderstanding**. As mentioned in **the original submission Sec. 4 last paragraph**, the textual strings are extracted by OCR. And we condition on the textual strings rather than text patches for layout generation. There is no background pattern information for the text input and consequently no such shortcut. We therefore appeal the reviewer’s re-evaluation about the “soundness” rating and final rating.

---

> > ### Author Response · Authors · 2023-11-16
> > **Continued Response to Reviewer EPYk**
> >
> > 5. **[Whether there are only text as foreground object in the dataset and all the experiments?]**
> >     - For the training and testing on the CLAY dataset, there are images as foreground objects for conditional layout generation, because CLAY contains bounding box ground truth for a variety of foreground elements beyond texts, e.g. icons, tool bars, advertisements, etc. The bounding box ground truth facilitates evaluation benchmarking including Layout FID, Layout KID, IoU, and DocSim which require bounding box ground truth as a reference. For the other datasets, there is no image as foreground for conditioning because there is no corresponding bounding box ground truth collected. We respectfully disagree if the reviewer leans to reject this submission mainly due to our dataset contribution. It **contradicts all the other reviewers’ strength points** about our dataset collection.
> >
> > 6. **[Why Crello dataset is not multi modal?]**
> >     - Crello dataset focuses on vector graphic designs. According to **their paper (CanvasVAE) Sec. 3.2 Paragraph 2, Fig. 3,5,6,7, and Table 1**, their text information is inadequate and is rendered as “TEXT TEXT ...” placeholders. If their text contents are provided, it will be a multimodal dataset candidate and we will change our comment on it accordingly.
> >
> > 7. **[Whether it's possible to apply this paper to the problem: "Towards Flexible Multi-modal Document Models"?]**
> >     - It is not straightforward to apply our design to the problem in “Towards Flexible Multi-modal Document Models”, because many of our technical components are layout-specific and not suitable for their non-layout attribute reconstruction. For example, our DETR architecture (originally for object detection) and our loss terms $L_\text{gIoU}$, $L_\text{overlap}$, $L_\text{misalign}$ are specified to generate/reconstruct bounding box parameters. It would be an independent paper’s scope to apply DETR + VAEGAN for their masked autoencoder problem formulation. But we will cite and discuss this literature in the next iteration.
> >
> > 8. **[Eq 6, should it be p_1^i?]**
> >     - Thank you for pointing out the typo. It will be corrected in the next iteration.
> >
> > 9. **[Show diversity of the generation]**
> >     - In fact, we have already discussed the diversity of generation. Due to the space limit, we had to mention it in **the original submission Appendix Sec. E Paragraph 2 and Fig. 8**.

---

### Official Review · Reviewer_Gw2u · 2023-10-29

**Soundness:** 2 fair
**Presentation:** 2 fair
**Contribution:** 2 fair
**Rating:** 5
**Confidence:** 4

**Summary:**

The paper studies graphic layout generation conditioned multimodal inputs, including background image, foreground image and text.

The main contribution is to adapt an exciting Transformer-based detector architecture as a content-conditioned layout generator and explore its training under different generative frameworks including GAN, VAE and VAE-GAN.

A new ad banner dataset with rich semantic annotations is created and will be released for the training and evaluation of generative models for graphic layouts.

**Strengths:**

1. The paper is studying an important problem. Conditioning layout generation models on rich contents will certainly make the models more practically useful.

2. The newly constructed banner dataset with detailed and rich annotations can be of value to the layout generation community.

3. The evaluation is extensive and the results look good.

**Weaknesses:**

1. The amount of technical contribution is small. While I appreciate the great effort that has been input into the work on building the system, testing different design choices and building the banner dataset, I think technical novelty and insight brought by the paper is limited. The whole work is more like constructing a working system by borrowing techniques from another domain directly (e.g., DETR) and combining components from other existing layout methods, e.g., (Kikuchi et al., 2021) and (Li et al., 2020), without any significant modification. Thus, the paper may not be of great interest to the ICLR audience, and perhaps fits better with more system-oriented conferences or journals.

2. The evaluation is insufficient. The paper is aimed at conditional layout generation. However, all the quantitative metrics as well as the user study only evaluate layout quality, and another important aspect of results is ignored — how well generated layouts match the input contents. Thus, an experiment on layout-content consistency is needed but is missing in the current paper.

**Questions:**

None

---

> ### Author Response · Authors · 2023-11-16
> **Response to Reviewer Gw2u**
>
> We thank all the reviewers for their constructive suggestions, which help improve the completeness of our submission. We are encouraged that the reviews are positive in the following five levels:
>
> * The paper is "**easy to follow**" (Reviewer t8xx).
> * The problem we are researching is “**important**” (Reviewer Gw2u) and our idea is “**interesting**” (Review j8Mq), “**effective**” (Review j8Mq), “**practically useful**” (Reviewer Gw2u), and is **one of our strengths** (Reviewer EPYk).
> * Our dataset contribution is “**useful**” (Reviewer j8Mq), “**valuable**” (Reviewer Gw2u),  and is **one of our strengths** (Reviewer EPYk, Reviewer t8xx).
> * Our evaluation is “**extensive**” (Reviewer Gw2u), and our results are “**SOTA**” (EPYk), “**effective**” (Reviewer j8Mq), “**excellent**” (Reviewer t8xx), and “**good**” (Reviewer Gw2u).
> * Our graphical system and user study are **one of the strengths** (Reviewer EPYk).
>
> We now address individual questions of **Reviewer Gw2u** below.
>
> 1. **[Technical novelty]**
>     - We understand this is a subjective judgement, and the debate would never end. We would rather argue using Prof. Michael Black’s Guide to Reviewers: Novelty in Science (https://medium.com/@black_51980/novelty-in-science-8f1fd1a0a143). We quote some:
>        - About the novelty of our connections between two established fields, multimodal conditioned layout generation and visual detection, Prof. Michael Black says “The novelty arose from the fact that nobody had put these ideas together before.” “Fortunately, these connections also turned out to be valuable, resulting in practical algorithms that were state of the art.” “To see the connections for the first time, before others saw them, was like breathing for the first time.” “The resulting paper embodies the translation of the idea into code, experiments, and text. In this translation, the beauty of the spark may be only dimly glimpsed. My request of reviewers is to try to imagine the darkness before the spark.”
>        - About the novelty of our simple and reasonable reuse of LayoutGAN++ and DETR architectures, Prof. Michael Black says “I value simplicity over unnecessary complexity; the simpler the better. Taking an existing network and replacing one thing is better science than concocting a whole new network just to make it look more complex.” “If a paper has a simple idea that works better than the state of the art, then it is most likely not trivial. The authors are onto something and the field will be interested.” “The inventive novelty was to have the idea in the first place. If it is easy to explain and obvious in hindsight, this in no way diminishes the creativity (and novelty) of the idea.”
>        - About the novelty of our newly-collected dataset, Prof. Michael Black says “Novelty (and value) come in many forms in papers. A new dataset can be novel if it does something no other dataset has done, even if all the methods used to generate the dataset are well known.”
>     - We therefore appeal the reviewer’s re-evaluation about the “contribution” rating and final rating.
>
> 2. **[All the quantitative metrics as well as the user study only evaluate layout quality]**
>     - This **contradicts your strength point** that “the evaluation is extensive”. Moreover, this is a **factual misunderstanding**. We do render foreground elements on top of background images according to the real/generated layouts. See **Fig. 1,3,7 in the original submission** for the rendered samples, and **Fig. 3 caption** for the rendering procedure. As a result, all of the Image FID, Image KID, and user study are based on the photorealistic real/rendered images, so as to consider the layout-content consistency. We will highlight this in the next iteration. We therefore appeal the reviewer’s re-evaluation about the “soundness” rating and final rating.

---

> > ### Comment · Reviewer_Gw2u · 2023-11-23
> >
> > For the second point in the authors' response, layout-content consistency in fact refers to how well generated layouts (or rendered images) match the input contents, which can also be referred to as input matching. The Image FID and KID only measure how well rendered images match real images, thus focusing on perceptual quality rather than layout-content consistency. Take the user study for example. In order to test layout-content consistency, the participants should be shown a pair of generated designs as well as their input condition, and asked to tell which better matches (or presents) the input condition, rather than just being asked to judge which looks better.

---

### Official Review · Reviewer_j8Mq · 2023-10-30

**Soundness:** 3 good
**Presentation:** 3 good
**Contribution:** 3 good
**Rating:** 6
**Confidence:** 4

**Summary:**

This paper focuses on the layout generation task by reformulating it as a detection problem. A transformer-based architecture, i.e., LayoutDETR, is proposed to detect reasonable locations, scales and spatial relations for elements in a layout. A new banner dataset is established with rich semantic annotation. The proposed solution is further integrated into a graphical system to scale up the layout generation process.

**Strengths:**

The idea of applying the visual detection framework for the layout generation task is interesting and effective. The collected could be useful for future research in the community. The experimental results show the effectiveness of the proposed method under six evaluation metrics.

**Weaknesses:**

1. The first contribution of this paper is that no existing methods can handle all those modalities at once. However, as shown in Table 1, Vinci can also use these modalities as conditions.
2. The computation cost analysis of the proposed solution is missing. Since the model contains a variety of input modalities, I was wondering about the computational cost and runtime analysis of the proposed method and existing works.
3. It would be better to show the diversity of the generated layouts and discuss the limitations of the proposed method.

**Questions:**

1. How to distinguish the foreground image and the background image if the background images are defined with arbitrary sizes?
2. Why Image FID that uses image features pre-trained on ImageNet could be used to evaluate the quality of the rendered graphic designs?

---

> ### Author Response · Authors · 2023-11-16
> **Response to Reviewer j8Mq**
>
> We thank all the reviewers for their constructive suggestions, which help improve the completeness of our submission. We are encouraged that the reviews are positive in the following five levels:
>
> * The paper is "**easy to follow**" (Reviewer t8xx).
> * The problem we are researching is “**important**” (Reviewer Gw2u) and our idea is “**interesting**” (Review j8Mq), “**effective**” (Review j8Mq), “**practically useful**” (Reviewer Gw2u), and is **one of our strengths** (Reviewer EPYk).
> * Our dataset contribution is “**useful**” (Reviewer j8Mq), “**valuable**” (Reviewer Gw2u),  and is **one of our strengths** (Reviewer EPYk, Reviewer t8xx).
> * Our evaluation is “**extensive**” (Reviewer Gw2u), and our results are “**SOTA**” (EPYk), “**effective**” (Reviewer j8Mq), “**excellent**” (Reviewer t8xx), and “**good**” (Reviewer Gw2u).
> * Our graphical system and user study are **one of the strengths** (Reviewer EPYk).
>
> We now address individual questions of **Reviewer j8Mq** below.
>
> 1. **[Vinci also handles all these modalities]**
>     - We have already discussed **the obvious limitations of Vinci in the original submission Sec. 2 second last paragraph**: “Vinci relies on a finite set of predefined layout candidates to choose background images from a pool of food and beverage domains. Their method is unable to design layouts conditioned on arbitrary backgrounds in open domains, natural or handcrafted, plain or cluttered, like ours.” We agree with the reviewer to lower our key in the next iteration.
>
> 2. **[Computation cost and runtime analysis]**
>     - In fact, we have already measured training and runtime computation costs. Due to the space limit, we had to report them in **the original submission Appendix Sec. B 2nd last paragraph**: “We train on 8 NVIDIA A100 GPUs for 110k iterations in 4 days. Inference is much more efficient: we load only G into a single NVIDIA A100 GPU and it consumes only 2.82GB of memory. It takes only 0.38 sec to generate a layout given foreground and background conditions.” We will manage to move this part to the main paper.
>     - Following the reviewer's suggestion, we also measure and summarize the space complexity and training/runtime time complexity of our model variants and all the baselines below. Although our training is relatively costly, it is just one-time. Our runway cost is one of the most efficient.
>       | Method            | #parameters (M) | Training on 8x A100 GPUs (hours) | Runtime on 1x A100 GPU (seconds per layout) |
> |-------------------|:---------------:|:-----------------------:|:-----------------------------------:|
> | LayoutGAN++       |       252       |            52           |                 0.38                |
> | READ              |       310       |            89           |                 0.38                |
> | Vinci             |       251       |            51           |                 0.36                |
> | LayoutTransformer |       190       |            36           |                 0.39                |
> | CGL-GAN           |        88       |            77           |                 0.40                |
> | ICVT              |        61       |            34           |                 0.44                |
> | LayoutDETR-GAN    |       270       |            96           |                 0.38                |
> | LayoutDETR-VAE    |       189       |            35           |                 0.38                |
> | LayoutDETR-VAEGAN |       351       |            99           |                 0.38                |
>
> 3. **[Show diversity of the generation]**
>     - In fact, we have already discussed the diversity of generation. Due to the space limit, we had to mention it in **the original submission Appendix Sec. E Paragraph 2 and Fig. 8**.
>
> 4. **[The limitation of the proposed method]**
>     - In fact, we have already discussed the limitation. Due to the space limit, we had to mention it in **the original submission Appendix Sec. F and Fig. 9**.
>
> 5. **[How to distinguish foreground and background images?]**
>     - For our ad banner dataset and CGL dataset, we intend not to distinguish foreground and background images as there is no foreground bounding box ground truth collected. All the visual contents except texts are regarded as the background image condition. On the other hand, because CLAY dataset has the ground truth annotation for all the foreground visual elements, all the training and testing on CLAY are conditioned on textual plus visual foreground inputs (e.g. icons).
>
> 6. **[Why is Image FID pretrained on ImageNet a useful metric?]**
>     - Image FID and Image KID are standard metrics to evaluate the photorealism between real and generated images. In our case, the real/generated images are rendered by overlaying foreground elements on top of background according to the real/generated layouts. Considering foreground and background images contain natural images, Image FID and Image KID are meaningful to measure the layout-content consistency.

---

### Meta-Review · Area_Chair_V26s · 2023-12-14

**Metareview:**

This paper presents LayoutDETR, a transformer-based architecture for layout generation reformulated as a detection problem, and introduces a new dataset with semantic annotations. The approach of applying a visual detection framework to layout generation is innovative, and the dataset is recognized as a valuable contribution to the field. The paper's clear presentation and its alignment with an important problem in the community are also commended.

However, there are significant areas that require major revisions. These include a more detailed elucidation of the novelty and technical contributions, particularly in differentiating the proposed methods from existing works. Additionally, the evaluation needs to be more comprehensive, addressing not only the quality of the layouts but also how well they correspond to the input content. Concerns about potential biases in the dataset and the overall methodology also need to be addressed thoroughly.

But the AC would like to note that the decision to reject the paper at this stage should be seen as an opportunity for further development and refinement. The innovative nature of the approach and the creation of a valuable dataset indicate a strong foundation for future contributions to the field. The authors are encouraged to take this feedback constructively and consider revising their work for a future conference, where the required comprehensive revisions can be more feasibly addressed.

**Justification For Why Not Higher Score:**

Save as above.

**Justification For Why Not Lower Score:**

N/A

---

### Decision · Program_Chairs · 2024-01-16

Reject